# Contributions of primary sources to submicron organic aerosols in Delhi, India

Sahil Bhandari[1], Zainab Arub[2], Gazala Habib[2], Joshua S. Apte[3,4], and Lea Hildebrandt Ruiz[5]

[1]Department of Mechanical Engineering, University of British Columbia, Vancouver, Canada
[2]Department of Civil Engineering, Indian Institute of Technology Delhi, New Delhi, India
[3]Department of Civil and Environmental Engineering, UC Berkeley, California, USA
[4]School of Public Health, UC Berkeley, California, USA
[5]McKetta Department of Chemical Engineering, The University of Texas at Austin, Texas, USA

*Correspondence to*: Lea Hildebrandt Ruiz (lhr@che.utexas.edu)

**Abstract.** Delhi, India experiences extremely high concentrations of primary organic aerosol (POA). Few prior source apportionment studies on Delhi have captured the influence of biomass burning (BBOA) and cooking (COA) on POA. In a companion paper, we develop a new method to conduct source apportionment resolved by time of day using the underlying approach of positive matrix factorization (PMF). We call this approach "time-of-day PMF" and statistically demonstrate the improvements in this approach over traditional PMF. Here, we quantify the contributions of BBOA, COA, and hydrocarbon-like organic aerosol (HOA) by applying positive matrix factorization (PMF) resolved by time-of-day on two seasons (winter and monsoon 2017) using organic aerosol measurements from an Aerosol Chemical Speciation Monitor (ACSM). We deploy the EPA PMF tool with the underlying Multilinear Engine (ME-2) as the PMF solver. We also conduct detailed uncertainty analysis for statistical validation of our results.

HOA is a major constituent of POA in both winter and monsoon. In addition to HOA, COA is found to be a major constituent of POA in monsoon and BBOA is found to be a major constituent of POA in the winter. Neither COA nor BBOA was resolved in the seasonal (not time-resolved) analysis. The COA mass spectral profiles (MS) are consistent with mass spectral profiles from Delhi and around the world, particularly resembling MS of heated cooking oils with a high *m/z* 41. The BBOA MS have a very prominent *m/z* 29 in addition to the characteristic peak at *m/z* 60, consistent with previous MS observed in Delhi and from wood burning sources. In addition to separating the POA, our technique also captures changes in MS profiles with the time of day, a unique feature among source apportionment approaches available. In addition to the primary factors, we separate 2–3 OOA components. When all factors are recombined to total POA and OOA, our results are consistent with seasonal PMF analysis conducted using EPA PMF. Results from this work can be used to better design policies that target relevant primary sources of organic aerosols in Delhi.

## 1 Introduction

Exposure to fine particulate matter (PM) poses significant health risks, especially in densely populated areas (Pope and Dockery, 2006; Apte et al., 2015). The Indian National Capital Region (Delhi NCR, India) is a rapidly growing urban agglomeration and encompasses the second most populated city in the world, with extremely high winter PM concentrations and frequent severe air pollution. According to a recent estimate, Delhi is the world's most polluted megacity, on track to also become the world's most populated megacity by 2028 (World Health Organization, 2018; United Nations, 2018).

Delhi has a long history of receptor-modeling studies focused on the quantity and composition of suspended particulate matter (Mitra and Sharma, 2002, Sharma et al., 2003; Mönkkönen et al., 2004, 2005a, b). Recent ambient studies have emphasized $PM_{2.5}$ and have attributed Delhi's aerosol pollution to vehicular traffic, fossil fuel combustion, and road dust (Srivastava et al., 2008; Tiwari et al., 2009, Pant and Harrison, 2012, Pant et al., 2015, 2016). Most of these studies identified vehicular emissions and fossil fuel combustion as prevalent factors contributing to the $PM_{2.5}$ pollution in Delhi. However, cooking and biomass burning have not been consistently identified as contributing sources in these ambient receptor modelling studies. This is not surprising since mass spectrometry-based studies have detected strong similarities in submicron aerosol mass spectra (MS) for traffic, cooking, and biomass burning, suggesting mixing of source signatures (Zhang et al., 2011). The lack of separation of these sources in receptor modelling studies suggests that traffic contributions to fine PM in Delhi are overestimated in such studies.

Filter-based receptor-modelling studies in Delhi are also marked by another limitation—only online mass-spectrometry studies have apportioned total organic PM (Bhandari et al., 2020; Tobler et al., 2020). Most filter-based studies have modelled components of organic PM such as organic carbon (OC), elemental carbon (EC), polycyclic aromatic hydrocarbons (PAHs), and water-soluble organic compounds (Pant et al., 2015; Sharma and Mandal, 2017). Some receptor modeling studies have resolved biomass burning as a separate factor and have identified cooking as contributing to fine PM. One such study studied 29 particle-phase organic compounds and a few elements in a molecular-marker-based Chemical Mass Balance (CMB) source apportionment study for the period 2001–2002 (Chowdhury et al., 2007). They identified biomass burning as the largest contributor to $PM_{2.5}$ in the winter of Delhi with a fractional contribution of ∼20%; vehicular emissions (gasoline, diesel) contributed 19% and coal burning contributed 14% at the Delhi site. For OC, they showed that traffic and coal combustion, and biomass burning contribute ∼15% each. A study focusing on $PM_1$ composition and source apportionment in Delhi for the colder months (Nov 2009–Mar 2010) attributed ∼7% to biomass burning and ∼4% to traffic; however, this study did not apportion organics (Jaiprakash et al., 2017). Another study conducted for a month in winter 2013–14 and two weeks in summer 2014 used pragmatic mass closure to attribute ∼23% of $PM_{2.5}$ to woodsmoke and ∼16% to traffic (Pant et al., 2015). The same study attributed ∼25% organic mass (OM) to traffic and ∼33% OM to woodsmoke in winter 2013–14 and attributed ~35% OM to traffic in summer 2014. Using EC, OC, other elemental analysis, and measurements of water-soluble inorganic ions measured over Jan 2013–May 2014, Sharma and Mandal (2017) apportioned ∼18% of $PM_{2.5}$ in Delhi to vehicular emissions, ∼13% to fossil fuel burning (coal, oil, and refuse burning), and ∼12% to biomass burning. The same

study attributed ∼55% OC to biomass burning and ∼35% OC to traffic. Another study made similar measurements for $PM_{2.5}$ over Nov 2013–Jun 2014 and identified vehicles, biomass burning, and coal and fly ash as contributing 25%, 28% and 5% in winter and 8%, 12%, and 26% in summer, respectively (Nagar et al., 2017). The study did not report apportionment results for OC separately. A four-year study (Jan 2013–Dec 2016) conducting seasonal PMF analyses on EC, OC, elemental analysis, and measurements of water-soluble inorganic ions reported $PM_{2.5}$ contributions of vehicular emissions as ∼20% and ∼17%, contributions of biomass burning as ∼19% and 15%, and fossil fuel combustion contributions of ∼9% and ∼9% in winter and monsoon 2013–2016, respectively (Jain et al., 2021). This study attributed ∼67% OC to vehicular emission and ∼10% to biomass burning in the winter season, and attributed ∼50% OC to vehicular emissions, ∼20% to biomass burning, and ∼8% to fossil fuel combustion in monsoon. Gadi and co-workers conducted PMF on 50–60 species of organics collected on filter samples between Dec 2016–Dec 2017 representing five classes of polar organic compounds in two studies (Gadi et al., 2019; Shivani et al., 2019). They measured $PM_{2.5}$ concentrations of 5 classes of polar organic compounds: alkanes, phthalates, PAHs, levoglucosan, and alkanoic acids. Their results indicate contributions of vehicular emissions (32–35%), biomass burning (27–30%), cooking emissions (16–17%), and plastic and waste burning (13%) to $PM_{2.5}$ in the National Capital Region (NCR) of Delhi.

Three large studies have reported data to state and federal agencies. One such study measured EC, OC, molecular markers such as polycyclic aromatic hydrocarbons (PAHs), elemental species, and water-soluble ions, and identified domestic sources (liquified petroleum gas (LPG), kerosene and wood combustion, diesel generator sets; 48—89%) and vehicular transport (6—22%) as key contributing sources to $PM_{2.5}$ across residential, curbside, and industrial locations averaged over the entire year of 2007 (except the monsoon season) (Central Pollution Control Board, 2010). Another such study measuring a similar set of species for the period 2013–2014 identified biomass burning and solid waste burning contributing about 24–34%, vehicular emissions contributing about 20–25%, and coal combustion (coal and fly ash) contributing about 5% to $PM_{2.5}$ in winter. In summer, soil and road dust (28%), and coal and fly ash (26%) were identified as the largest sources, along with biomass burning and solid waste burning (19%), and vehicular traffic (9%) (IIT Kanpur, 2016). The third study measured EC, OC, elements, and ions and identified vehicular emissions and biomass burning contributing 23% and 22% respectively in winter and 18% and 15% in summer to $PM_{2.5}$ concentrations in Delhi for the year 2016 (ARAI and TERI, 2018). None of these three studies reported contributions of different sources to OC or OM separately.

Bottom-up studies such as those on personal exposure and using source-oriented modeling have recognized the high exposure to residential energy emissions from sources such as cooking and heating and associated biomass burning emissions and their significant impact on human health (Pant et al., 2017; Apte and Pant, 2019, and references therein). Emissions inventories developed for Delhi and other regions across South Asia have shown sources such as transport, industry, dust, household solid fuel use, and biomass and waste burning to contribute substantially to $PM_{2.5}$ (Guttikunda and Calori, 2013; IIT Kanpur, 2016; ARAI and TERI, 2018; Conibear et al., 2018; GBD MAPS Working Group, 2018; Guo et al., 2018). Two modeling studies based on EDGAR and other emissions inventories for the year 2015 estimated local residential and industrial sources accounting for most of the primary $PM_{2.5}$ concentrations in Delhi. Residential sources were separated into ∼42% as

associated with biomass burning and ~58% as associated with fossil-fuels (Guo et al., 2017, 2019). The study attributed ~12 $\mu gm^{-3}$ of primary organic carbon (POC) to residential fossil fuel combustion and ~8 $\mu gm^{-3}$ to residential biomass burning. ARAI and TERI (2018) estimated $PM_{2.5}$ contributions of 28% and 17% from vehicular transport and 14% and 15% from residential emissions and agricultural burning for winter and summer seasons for Delhi in the year 2016. Rooney et al (2019) used a similar emissions inventory and estimated residential emissions to contribute ~10% to the anthropogenic $PM_{2.5}$ burden in New Delhi. Neither ARAI and TERI (2018) nor Rooney (2019) reported apportionment of OC or OM.

In the last few years, few receptor-modeling studies have used high time-resolution online mass spectrometry instrumentation to conduct source apportionment on Delhi's organic aerosols (OA) (Bhandari et al, 2020; Tobler et al., 2020; Cash et al., 2021; Reyes-Villegas et al., 2021; Shukla et al., 2021; Lalchandani et al., 2021). Based on NR- $PM_{2.5}$ ACSM data collected, one study identified hydrocarbon-like organic aerosol (HOA) and solid-fuel combustion organic aerosol (SFC-OA) as the primary PMF factors, with HOA contributing ~15% and ~17%, and SFC-OA contributing ~25% and ~17% to total organic NR- $PM_{2.5}$ in winter 2018 and summer 2018 (the month of May), respectively (Tobler et al., 2020). They suspect the SFC-OA to comprise of contributions from domestic heating, coal-based cooking, and open-fire activities, including many types of biomass burning and waste combustion. In the other study, biomass burning organic aerosol (BBOA) was separated as a factor only in spring 2018, contributing ~15%, and HOA contributed ~16% to the NR- $PM_1$ OA burden in Delhi (Bhandari et al., 2020). The other studies used relaxed factor identification criteria, and identify multiple SFC-OA and HOA factors across multiple urban and urban-downwind sites in Delhi (Cash et al., 2021; Reyes-Villegas et al., 2021; Shukla et al., 2021; Lalchandani et al., 2021). In urban areas of Delhi, these studies find that in winters, HOA accounts for ~20–40% fine OA, and SFC-OA, BBOA, and COA account for 20–30% fine OA, with higher SFC-OA downwind (Reyes-Villegas et al., 2021; Lalchandani et al., 2021). In summer and monsoon, HOA accounts for ~10–30% fine OA, and SFC-OA, BBOA, and COA account for ~15–30% fine OA (Cash et al., 2021; Shukla et al., 2021). Overall, online mass spectrometry studies have reported limited measurements of biomass burning contributions and have not always resolved cooking organic aerosol (COA) as a factor in Delhi, despite its ubiquity (Tobler et al., 2020; Shukla et al., 2021; Lalchandani et al., 2021).

In recent years, several multi-institutional studies have been initiated for Delhi on sources, emissions, and atmospheric dynamics of fine aerosols, as well as their relationship to health impacts (NASA JPL, 2020; Venkataraman et al., 2020; NERC–MRC-MoES-DBT Atmospheric Pollution and Human Health program, 2021). However, few clean air action plans in India utilize results from studies on source contributions to formulate action plans (Ganguly et al., 2020). This limitation of policy instruments needs to be addressed to allow regulators to better hold priority polluting sources to account. Incorporation of source contribution studies as a part of action plans also allows systematic evaluation of both long-term and policy-specific changes. As an example, recent national initiatives promoting use of cleaner fuels may not have achieved their intended targets, and source apportionment studies will assist with quantifying policy impacts (Kar et al., 2020). We believe that the inability to separate prominent sources such as cooking and biomass burning in source apportionment studies on fine aerosols in Delhi likely limits the confidence policymakers place in source apportionment tools.

This paper improves upon the seasonal source apportionment previously employed in Delhi (Bhandari et al., 2020). The Delhi Aerosol Supersite (DAS) study provides long-term chemical characterization of ambient submicron aerosol in Delhi, with near-continuous online measurements of aerosol composition (Gani et al., 2019; Arub et al., 2020; Bhandari et al., 2020; Gani et al., 2020; Patel et al., 2021a; Patel et al., 2021b). In that study, PMF was conducted on six seasons of highly time-resolved speciated non-refractory submicron aerosol (NR- $PM_1$) organic (Org) mass spectrometer data from an Aerosol Chemical Speciation Monitor (ACSM) in the PMF receptor model at a time resolution of 5–6 min. Then, we deployed the IGOR PET tool on seasonal datasets and 2–3 PMF factors were extracted (Bhandari et al., 2020). In all but one season, we could not resolve primary organic aerosol (POA) into component primary factors. In a companion paper, we have developed a new technique to separate POA into component primary factors called the "time-of-day PMF" approach (Bhandari et al., 2022).

Here, we report on time-of-day PMF conducted on ACSM organic aerosol data from the two seasons of winter and monsoon 2017—collected as a part of the Delhi Aerosol Supersite (DAS) study. Since we resolve the dataset by time-of-day, thus factor MS is expected to vary in these time-of-day windows. The two seasons of winter and monsoon are selected for this analysis as they capture two extremes in seasonal concentrations, precipitation, and meteorology, especially in terms of temperature, ventilation coefficient, wind direction, and wind speed (Fig. S1, Tables S1, S2). In addition, winter experiences extremely high organic and inorganic concentrations and high pollution episodes dominated by primary emissions (Gani et al., 2019; Bhandari et al., 2020). We used the EPA PMF tool to apply constraints, extract a larger number of factors, and quantify errors in PMF solutions.

## 2 Methods

### 2.1 Statistical basis of approach

ME-2 is a multilinear unmixing model that can be used to perform bilinear deconvolution of a measured mass spectral matrix (**X**) into the product of positively constrained mass spectral profiles (**F**) and their corresponding time series (**G**) (Eq. 1). **E** corresponds to the data residual not fit by the model. The elemental notation of Eq. 1 is shown in Eq. 2. Given that time series (TS) and mass spectra (MS) are deconvoluted, the model mass spectral profiles are assumed to remain constant in time. The mass balance equation underlying the bilinear implementation of the factor analytical model and the optimization problem in the EPA PMF tool can be represented as shown in Eqs. 1–3.

$$\mathrm{X = GF + E} \tag{1}$$

$$x_{ij} = \sum_{p=1}^{n} g_{ip} \, . f_{pj} \; + e_{ij} \tag{2}$$

$$Min_{\mathbf{F,G}} Q = \sum_{i=1}^{m} \sum_{j=1}^{n} \left(e_{ij}/\sigma_{ij}\right)^2 \tag{3}$$

To derive factor time series and mass spectra in an iterative fitting process, ME-2 lowers residual by minimizing the quality of fit parameter $Q$ using the gradient approach (Norris et al., 2014). Here, we do not expect the norm of the actual error matrix

to be zero but instead close to the ACSM measured uncertainty (an element of the measured uncertainty is represented as $\sigma_{ij}$ in Eq. 3). The quality of fit parameter corresponding to this uncertainty is called $Qexp$ (Ulbrich et al., 2009). Usually, PMF solutions start from very high $Q/Qexp$ and converge to 1 as we add more factors.

A key limitation of PMF is that it assumes constant MS profiles, even though source signatures can change over the course of the day. MS also changes due to reaction chemistry, gas-particle partitioning, effect of changing meteorology and trajectory sampling, and human activity patterns, processes that vary diurnally (Lelieveld et al., 1991; Abdullahi et al., 2013; Zhang et al., 2013; Venturini et al., 2014; Park et al., 2019; Pauraite et al., 2019; Crippa et al., 2020; Dai et al., 2020; Xu et al., 2020). To address this limitation of constant MS profiles, we used two alternative approaches for conducting PMF. In one approach, we applied PMF by splitting the data into six 4 hour time windows each day to illustrate the use of our time-of-day PMF method. We also conduct seasonal PMF runs for winter and monsoon 2017. We refer to the traditional seasonal organic MS-based PMF analysis results as "seasonal PMF" and time-of-day organic MS-based PMF analysis results as "time-of-day PMF" results in the paper. Thus, we conducted 14 PMF runs in total (two seasons: 2 seasonal PMF and 12 (two times six) time-of-day PMF). Details of the statistical basis of the technique can be found in the companion paper (Bhandari et al., 2022). To refer to PMF runs corresponding to specific time windows, we use the nomenclature "Season" + "Year" + "Period" style in the format "SYYTTTT" (Table S3). For example, W171115 corresponds to the 1100–1500 hours of Winter 2017.

### 2.2 Sampling site and measurements

As a part of the DAS campaign, an ACSM (Aerodyne Research, Billerica, MA) was operated at ∼1 min time resolution in a temperature-controlled laboratory on the top floor of a four-story building at IIT Delhi (Ng et al., 2011b). Full details of sampling site, instrument setup, operating procedures, calibrations, and data processing are described in a separate publication (Gani et al., 2019). We collected the data used in this paper in winter (January-February 2017) and monsoon (July-September 2017). Definition of the seasons comes from the Indian National Science Academy (2018) (Table 2 from Bhandari et al., 2020). Diurnal plots of meteorological variables are shown in Fig. S1. We conducted separate PMF analysis for 6 time-of-day periods in both winter and monsoon, with our data categorized into 12 time-of-day periods over these two seasons, together with two seasonal PMF runs (Table S3). We used the dataset obtained by averaging every five consecutive measurements. We selected organic spectral data at a specific set of m⁄z values between m⁄z 12 and m⁄z 120. This approach is the commonly used approach, and the reasons for the selection of the specific set of m⁄z values have been described previously (Zhang et al., 2005). Spring, summer, and autumn (mid-September to November) periods are not included in the analysis here; but seasonal PMF analysis has been presented in previous publications (Bhandari et al., 2020; Patel et al., 2021a).

### 2.3 PMF (ME-2) Analysis

The EPA PMF v5.0 tool was used to conduct ME-2 analysis on the dataset and interpret the results (Norris et al., 2014). Further details on the statistical basis of ME-2 are available elsewhere (Paatero et al., 1999; Paatero et al., 2002). For the base run, the

iterative PMF technique does not make any assumptions for source or time profiles. If factors extracted in the base run are not clearly associated with a source type but suggestive of presence of mixing of specific sources, constraints are applied on the factors in the base run to extract cleaner source profiles (Brown et al., 2012; Brown et al., 2015). An R package was developed to automate the process of data analysis of EPA PMF outputs (R Core Team, 2019). We readjusted the results from PMF analysis to account for underestimation of factor mass based on the selected m/z values only. To account for particle losses, we applied transmission and collection efficiencies after conducting PMF analysis (Gani et al., 2019). Additional details of the R code and criteria for factor selection have been discussed in detail in a separate publication (Bhandari et al., 2022).

Details of solution identification and criteria for factor selection can be found in the companion paper (Sect. S1 in Bhandari et al., 2022). Selection of PMF factors and application of factor constraints are discussed in the Supplement (Sect. S2; Tables S3–S6). We also utilized error estimation (BS, DISP, BS-DISP) and constraints in the EPA PMF tool for factor selection (Tables S7–S10). An updated method was used for selection of bootstrap block size (Table S7; Sect. S2 in Bhandari et al., 2022). We used the Pearson correlation coefficient (Pearson R) for mass spectral data and Spearman correlation coefficient (Spearman R) for time series patterns. This differentiation was recommended in the peer review of Ulbrich and co-workers (2009), due to the limitations of Pearson R for slowly varying time series concentrations.

### 2.4 Application of the hybrid MLR-PMF approach to the primary PMF components

Sometimes, positive matrix factorization does not resolve factors present in small amounts. Ulbrich and co-workers (2009) suggested 5% of the organic aerosol factor mass as a lower limit of detection for factor retrievals of Q-AMS measurements for PMF factors that have similar MS and TS structure. This is indeed the case for primary source mass spectra for traffic (HOA), cooking (COA), and biomass burning (BBOA) (Zhang et al, 2011). Thus, organic aerosol PMF solutions are associated with some mixing in factor profiles, especially when PMF is not able to separate all component PMF factors. For this work, we assumed that primary factors obtained in PMF can be expressed as a sum of HOA, COA, and BBOA. For periods where we are unable to extract all three factors, we applied a multilinear regression (MLR) approach on primary factor MS to extract the three factors. Hereafter, we refer to the approach as "hybrid MLR-PMF". Similar approaches using the Multilinear Engine or PMF combined with multilinear regression have been deployed previously (Lin et al., 2017; Cash et al., 2021). A key limitation of this approach is that it assumes collinear time series patterns in the time windows for the extracted factors. Details of the approach are discussed in the Supplement (Sect. S3). Here, we applied this approach on time-of-day PMF results. For 8 time-of-day PMF runs with two primary PMF factors, this approach reproduced the results of PMF with mixing of ≤5% OA, pointing to the validity of the approach. For the other 4 time-of-day PMF runs, we used this approach as an attempt at extracting all three PMF factors. However, in all cases, we extract substantial concentrations for at most two primary factors (Tables 1 and 2). These four periods are associated with low fractional contributions of POA ≤~30% (Tables S11 and S12).

## 3 Results and discussion

In the companion paper, we introduce the time-of-day PMF analysis as an approach to account for changing MS profiles over the day (Bhandari et al., 2022). In this paper, we focus on the components of organic aerosol obtained using time-of-day PMF and compare them to the seasonal PMF approach. We report average seasonal concentrations of time-of-day PMF factors for winter 2017 in Table 1 and for monsoon 2017 in Table 2. In both seasons, winter and monsoon, the 6 time-of-day periods are marked by 3 distinct transitions in total concentrations, one at midday (1100 hours), one at night (1900 hours), and one in the early morning (0300 hours), reflective of the changing ventilation and other meteorological variables, and sources influencing the bracketed 8-hour windows (Tables 1 and 2; Fig. S1). Dividing time series into different time-of-day periods for applying PMF allows further separation of primary organic aerosol (POA) into component factors hydrocarbon-like organic aerosol (HOA), biomass burning organic aerosol (BBOA), and cooking organic aerosol (COA)—a result likely arising from the ability to extract variable mass spectral profiles over the day (Bhandari et al., 2022). On an average, we observe that winter POA mass comprises of 45% HOA and 55% BBOA, and monsoon POA is a combination of 49% HOA and 51% COA (Tables S11 and S12). In winter, we separated BBOA or BBOA-like factors in all periods (four separations based on PMF, two separations based on hybrid MLR-PMF) but did not separate COA (Table 1). In monsoon, we separated COA or COA-like factors in all periods (four separations based on PMF, two separations based on hybrid MLR-PMF) but did not separate BBOA above detection limits (Table 2). We also separated HOA in all time-of-day periods in winter and monsoon (four separations based on PMF, two separations based on hybrid MLR-PMF) (Tables 1 and 2). Figure 1a–d shows the time series of the POA (HOA+BBOA+COA), and oxygenated organic aerosol (OOA) factors for winter and monsoon of 2017. The interplay of sources, meteorology, and photochemistry results in sharp variations in PMF factor concentrations across seasons. Clearly, POA concentrations exhibit larger variability than OOA concentrations. We show the diurnal variability of the PMF factors in different seasons in Figs. 2 and 3. Fractional contributions of PMF factors are shown in Tables S11 and S12. Our results show that the time series (TS) concentrations of time-of-day PMF factors are broadly consistent with seasonal PMF factors (Sect. 3.3, Figs. 4 and 7). Mass spectra (MS) of these factors are discussed in detail in Sect. 3.3 (Figs. 5–6, 8–9). Quantitative contributions at key *m/zs* are shown in Tables S15–S26.

**Table 1 Average seasonal concentrations of time-of-day PMF factors for winter 2017 (in μgm$^{-3}$)**

| Period | HOA | BBOA | COA | OOA | Total |
|---|---|---|---|---|---|
| W171115[a] | 5 | 18 | 0 | 47 | 70 |
| W171519 | 14 | 10 | 0 | 35 | 58 |
| W171923 | 32 | 44 | 0 | 66 | 142 |
| W172303 | 36 | 36 | 0 | 71 | 143 |
| W170307[b] | 22 | 14 | 0 | 80 | 117 |
| W170711 | 18 | 18 | 0 | 83 | 119 |

[a]For W171115, we were able to separate POA into solid fuel combustion organic aerosol (SFC-OA) and BBOA, not HOA and BBOA. We used hybrid MLR-PMF to apportion SFC-OA to HOA and BBOA. The entry in W171115 BBOA contains

contributions from the PMF BBOA factor as well as a BBOA contribution from hybrid MLR-PMF-based SFC-OA apportionment. [b]Hybrid MLR-PMF-based results for W170307

**Table 2 Average seasonal concentrations of time-of-day PMF factors for monsoon 2017 (in μgm$^{-3}$)**

| Period | HOA | BBOA | COA | OOA | Total |
|---|---|---|---|---|---|
| M171115[a] | 1.3 | 0.1[b] | 2.6 | 17 | 21 |
| M171519 | 3.7 | 0 | 1.3 | 13 | 18 |
| M171923 | 5.9 | 0 | 6.5 | 18 | 30 |
| M172303 | 7.7 | 0 | 4.4 | 18 | 30 |
| M170307 | 4.5 | 0 | 3.4 | 15 | 23 |
| M170711[a] | 0.4 | 0 | 3.9 | 20 | 24 |

[a]Hybrid MLR-PMF-based results for M171115 and M170711 [b]Below organic detection limit in the ACSM (Ng et al., 2011b)

In Sect. 3.1, we discuss the separation of primary factors in different seasons. We also compare the contributions of the different primary organic aerosol components to Delhi's submicron aerosols with literature estimates for fine aerosols. In Sect. 3.2, we discuss the mass spectral profiles and diurnal time series patterns of the primary PMF factors across the time-of-day windows. We observe separation of the secondary PMF factors into local and regional OOA with evidence of some mixing, in line with recent observations (Drosatou et al., 2019; Bhandari et al., 2022). Thus, we do not discuss OOA component factors in detail. In Sect. 3.3, we compare the TS contributions and MS of total POA and OOA from time-of-day PMF and seasonal PMF.

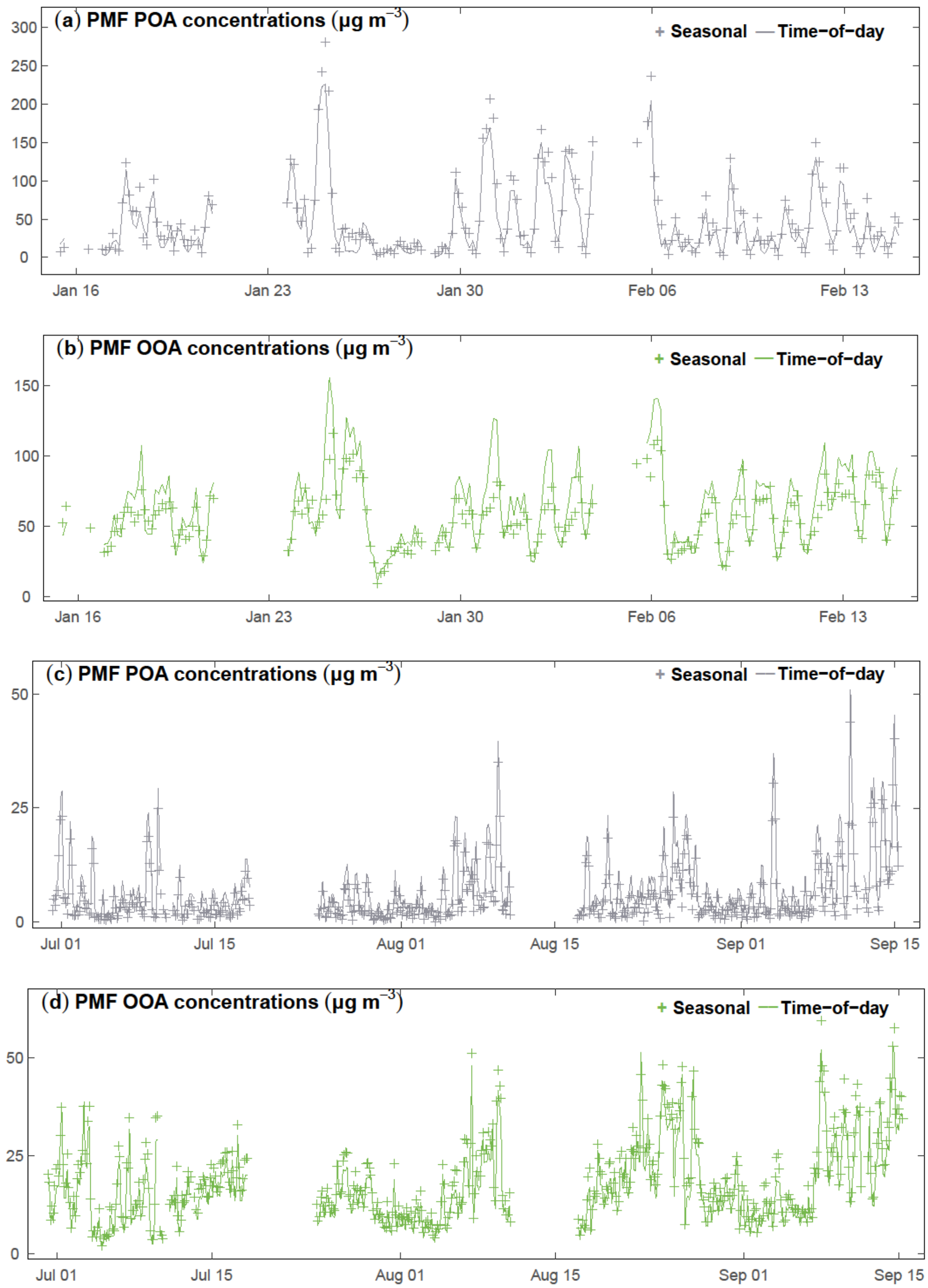

**Figure 1 shows the 4 h averaged concentration time series (TS) of time-of-day PMF (lines) and seasonal PMF (+) primary organic aerosol (POA) and oxygenated organic aerosol (OOA) factors in periods: (a)-(b) winter 2017, and (c)-(d) monsoon 2017 (in µgm$^{-3}$). POA factor concentrations show stronger seasonal variations than OOA.**

### 3.1 Separation of primary factors

In winter, for four of the six time-of-day PMF runs, we obtained two POA factors—a hydrocarbon-like organic aerosol (HOA) and a biomass-burning organic aerosol (BBOA) factor, and two–three oxidized organic aerosol (OOA) factors (Table S3). The two time-of-day periods in winter with a different set of PMF factors are W170307 and W171115. For W170307, PMF results in a single POA factor (a mixed HOA-BBOA factor) and two OOA factors. For W171115, PMF separates a solid-fuel combustion (SFC-OA) factor and a BBOA factor, along with two OOA factors. Application of the hybrid MLR-PMF approach separated W170307 POA and W171115 SFC-OA into HOA and BBOA factors. In monsoon, for four of the six time-of-day PMF runs, we obtained two POA factors—a hydrocarbon-like organic aerosol (HOA) and a cooking organic aerosol (COA) factor, and two OOA factors. For M170711 and M171115, PMF resulted in a single POA factor (a mixed COA-HOA factor) and two OOA factors. Application of the hybrid MLR-PMF approach separated M170711 and M171115 POA into HOA and COA factors. Concentrations of the HOA, BBOA, and COA factors are shown in Tables 1 and 2.

#### 3.1.1 Comparison to other organic apportionment studies

Recent studies have reported contributions of traffic-, cooking-, and biomass burning-related factors to $PM_1$ and $PM_{2.5}$ concentrations in Delhi. Here, we show that our results obtained using the time-of-day PMF approach are broadly consistent with those studies.

In winter, we observe average contributions of HOA as 16% and BBOA as 19% to total organic non-refractory submicron aerosols (NR-$PM_1$) (and no detected contributions of cooking), in line with a recent online mass spectrometer deployment for NR-$PM_{2.5}$ (Tobler et al., 2020: HOA ~15%, SFC-OA~25%; Lalchandani et al., 2021: HOA ~20%, SFC-OA~30%; Table S11). There are large differences in comparison to another study (Reyes-Villegas et al., 2021: HOA~40%, BBOA~12%, COA~8%). However, this study conducted ground-based measurements (~3m high), within 50m of a major road. This siting likely resulted in high contributions of HOA to OA in this study. The observation of similar or larger contributions of biomass burning than traffic to organics in the winter of Delhi is consistent with several filter-based receptor modeling studies as well (Chowdhury et al., 2007; Pant et al, 2015; IIT Kanpur, 2016; Sharma and Mandal, 2017; Jaiprakash et al., 2017). In monsoon, we observe ~14% organic NR-$PM_1$ attributable to HOA and ~15% attributable to COA (no detected contributions of biomass burning) (Table S12). These results are in line with the online mass spectrometry study of Tobler and co-workers (2020; May 2018: NR-$PM_{2.5}$ contributions of HOA (17%) and SFC-OA (17%)). However, there are large differences in comparison to another study, particularly in HOA contributions (Cash et al., 2021: HOA~30%, SFC-OA~11%, COA~6%). However, this study conducted ground-based measurements (~3m high), within 50m of multiple traffic sources (railways, major road). This siting likely resulted in high contributions of HOA to OA in that study.

### 3.1.2 Comparison of organic apportionment to fine PM apportionment

Our results show that winter concentrations are higher than monsoon, and other seasons generally experience intermediate concentrations (Bhandari et al., 2020; Patel et al., 2021a). Thus, assuming that the fractional contributions of HOA, BBOA, and COA are similar in other seasons, we expect annual contributions to primary organics from biomass burning and cooking to be larger than or comparable to traffic. These results are in line with multiple receptor modeling and source-oriented modeling studies (Nagar et al., 2017; ARAI and TERI, 2018; Shivani et al., 2019; Jain et al., 2021). Assuming 50% of biomass burning and cooking contributions are coming from residential use, and low biogenic fine particle concentrations, we expect residential emissions contributing to ~10% of anthropogenic $PM_{2.5}$ in Delhi, in line with another source-oriented modeling study (Rooney et al., 2019). More broadly, the importance of local sources such as traffic, cooking, and biomass burning in Delhi is consistent with other studies (Guttikunda and Calori, 2013; Guo et al., 2017). These results suggest the consistency of recent top-down and bottom-up approaches and indicate high contributions of non-vehicular emissions to primary PM.

It is also important to point out that several receptor and source-oriented modeling studies identified large contributions of fossil-fuel combustion, separate from vehicular emissions, and attributed these contributions primarily to coal and fly ash emissions (Chowdhury et al., 2007; IIT Kanpur, 2016; Nagar et al., 2017; Sharma et al., 2017; Jain et al., 2021). No online mass spectrometry receptor modeling study in India has separated coal combustion organic aerosol as a PMF factor; however, coal combustion organic aerosol has been separated as a factor elsewhere (Dall'Osto et al., 2013; Hu et al., 2013; Wang et al., 2015; Sun et al., 2016; Zhu et al., 2018). We suspect that coal combustion OA is detected at our site at low concentrations. Similarly, emissions from garbage burning and emissions from brick kilns likely contribute at our site as well and have been identified as important contributing sources in a South Asian nation in the NAMASTE campaign (Misra et al., 2020; Werden et al., 2020; DeCarlo et al., 2021). Future studies could utilize these measurements to separate contributions of coal combustion organic aerosol, garbage burning, and emissions from brick kilns.

### 3.1.3 Variations in contributions across the day

To better understand the variations of different PMF factors in the two seasons, we plot diurnal patterns of average factor concentrations in the two seasons in Figs. 2 and 3. To allow an inter-seasonal comparison, we used the 1-D volatility basis set (VBS) approach to adjust for temperature (T) and ventilation coefficient (VC). Details of application of VBS are in the Supplement (Sect. S4).

Figure 2 here shows diurnal patterns of primary factors in monsoon. In dotted lines, we have also plotted winter BBOA adjusted for monsoon T and VC. Hereafter, we call this "winter-to-monsoon BBOA". If the difference in concentrations was due to partitioning and meteorology only, then the "winter-to-monsoon" concentrations should be comparable to the monsoon concentrations of the same PMF factor. We observe that time-of-day PMF generates higher primary OA concentrations than seasonal PMF in all time-of-day periods in monsoon 2017. Detailed comparison of POA and OOA factors in time-of-day PMF and seasonal PMF is shown in Sect. 3.3. Also, the "winter-to-monsoon" BBOA concentrations between

1700–0300 hours are higher than total POA concentrations in monsoon, highlighting the large seasonal disparity in PM concentrations in Delhi. We observe higher contributions of monsoon HOA in the early morning hours and late at night, in line with the heavy-duty traffic in the early morning hours and high light-duty traffic congestion at night on major traffic corridors (Mishra et al., 2019). We observe high contributions of monsoon cooking at cooking periods throughout the day. Overall, we observe that HOA (~49%) and COA (~51%) contribute almost equally to the primary organic aerosol burden in monsoon. Further, BBOA contributes only minimally. The monsoon HOA and COA concentrations are likely influenced by precipitation, and average concentrations might be lower by as much as a factor of 2 (Gani et al., 2019; Fig. S1). However, a detailed effect of precipitation on organic concentrations has not been conducted in this work. Our results suggest that the inability to separate BBOA in monsoon, particularly in the middle of the day (0900–1700 hours), can be attributed to the volatility of BBOA, as can be seen from the low BBOA concentrations estimated from hybrid MLR-PMF and winter-to-monsoon BBOA. While winter-to-monsoon BBOA is large at other hours, those BBOA concentrations are likely associated exclusively with winter night-time space heating. Thus, the lack of BBOA concentrations in monsoon are due to absence of sources (e.g. residential heating) as well as the relatively high volatility of BBOA, which would result in near-zero concentrations during summer daytime temperatures.

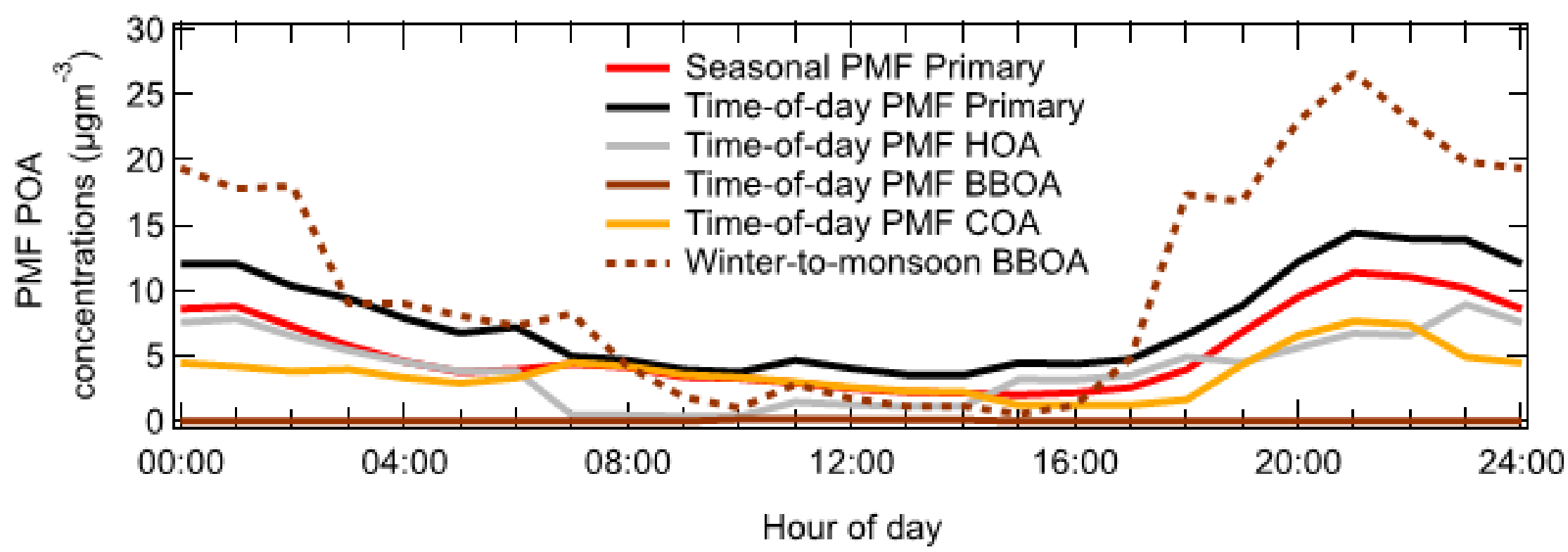

**Figure 2 Diurnal patterns of seasonally representative hourly averaged primary PMF factors in monsoon 2017. Traffic and cooking contribute almost equally whereas biomass burning has negligible contributions.**

Figure 3 shows the diurnal concentrations of primary factors in winter 2017. In dotted lines, we have also plotted monsoon COA and monsoon HOA adjusted for winter T and VC, hereafter referred to as "monsoon-to-winter COA" and "monsoon-to-winter HOA" respectively. We observe that time-of-day PMF generates higher primary OA concentrations than seasonal PMF in the middle of the day (1100–1900 hours). Detailed comparison of POA and OOA factors in time-of-day PMF and seasonal PMF is shown in Sect. 3.3. Also, HOA concentrations in winter are higher than monsoon-to-winter HOA, particularly in the early morning hours, suggesting contributions of winter-only sources to HOA. We discuss the plethora of sources contributing to winter HOA in Sect. 3.2. Like monsoon, traffic contributions are largest early in the morning and at night. Winter BBOA concentrations are comparable to winter HOA but increase particularly in the early morning hours, late at night, and in the middle of the day, likely due to the use of biomass burning for winter night-time space heating and for

solid-fuel combustion. Overall, traffic (45%) and biomass burning (55%) contribute nearly equally to the primary organic aerosol burden in winter. Further, COA contributes only minimally. Our results suggest that the inability to separate COA in winter can be attributed to its low concentrations compared to other POA factors, even though the COA sources might be the same as monsoon. Thus, our analysis explains the missing COA concentrations in winter when using PMF.

Diurnal concentrations of winter BBOA and monsoon COA are strongly correlated (Pearson R~0.89, Fig. S2). Indeed, recent work suggests that the use of wood and biomass combustion is primarily limited to use as cooking fuel in monsoon and additionally for heating in winter (Fu et al., 2010; Yadav et al., 2013; Pant et al., 2015, Rooney et al., 2019). The identification of factors such as SFC-OA associated with cooking periods, similar sources for COA and BBOA, similarity in diurnal patterns, and the selective separation of COA and BBOA as PMF factors in monsoon and winter respectively suggests interplay between COA and BBOA. Deployment of higher-resolution instrumentation can help address the reasons behind this interplay in more detail.

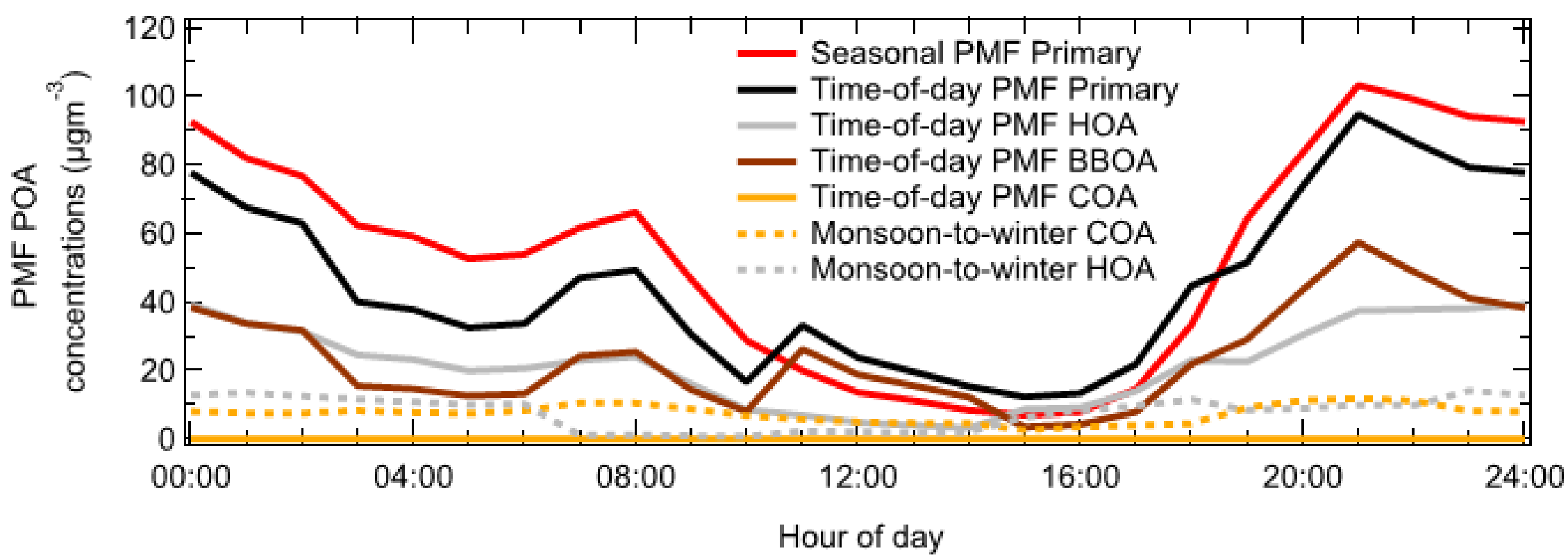

**Figure 3 Diurnal patterns of seasonally representative hourly averaged primary PMF factors in winter 2017. Traffic and biomass burning contribute almost equally whereas cooking has negligible contributions.**

### 3.2 Primary factor MS and TS

Here, we discuss the mass spectral profiles and time-series patterns of primary factors from the six periods of time-of-day PMF for both seasons of winter and monsoon. Depending on the time-of-day and season, PMF suggests a varying influence of HOA, BBOA, and COA (Tables 1 and 2). Diurnal patterns of mean and median concentrations of PMF factors in winter and monsoon are shown in Figs. 4 and 7.

The variability in MS characterizes the detected PMF factors (Zhang et al., 2011). The ratio of contributions at *m/z* 43 to *m/z* 44 is typically considered as a marker for the state of oxidation of the MS; the lower the value, the more oxidized is the detected MS (Ng et al., 2010). High contributions at *m/z* 57 are associated with high influence of traffic emissions (Ng et al., 2011a). High contributions at *m/z* 41 and high ratio of contributions of *m/z* 41 to *m/z* 43 are a characteristic feature of COA from heated cooking oils, especially in Asian cooking (Allan et al., 2010; He et al., 2010; Liu et al., 2018; Zheng et al., 2020). The ratio of contributions at *m/z* 55 to *m/z* 57 has been identified as a marker for the presence of cooking organic aerosol

(Robinson et al., 2018). Contributions at *m/z* 29, *m/z* 60, and *m/z* 73 are strongly influenced by wood and biomass burning (Bahreini et al., 2005; Schneider et al., 2006; Zhang et al., 2011). Mass spectral profiles for the HOA, BBOA, and COA factor MS in winter and monsoon are shown in Figs. 5–6 and 8–9. The factors representing traffic contributions (HOA) have consistently high correlations with hydrocarbon-like organic aerosol (Figs. S3 and S4; Pearson R>0.9). Also, in all periods, at least one of the separated factors resembles biomass burning organic aerosol in winter (Fig. S5, Pearson R>0.8) and cooking organic aerosol in monsoon (Fig. S6, Pearson R>0.7).

### 3.2.1 Winter 2017

In winter, we separated BBOA or BBOA-like factors in all periods (five separations based on PMF, one separation based on hybrid MLR-PMF) but did not separate cooking organic aerosol (Table S3). We also separated HOA or HOA-like factors in all time-of-day periods in winter (four separations based on PMF, two separations based on hybrid MLR-PMF). At winter midday, PMF did not separate an HOA factor but separated an SFC-OA factor. Details of this SFC-OA factor have been discussed in a companion publication (Bhandari et al., 2022). Similar SFC-OA factors have been reported in literature (Tobler et al., 2020; correlation at all *m/z*s but *m/z* 44, Pearson R>0.95; Fig. S7). Here, we discuss the MS and TS patterns of separated HOA and BBOA factors in time-of-day periods of winter 2017.

<u>HOA MS and TS</u>

We show that the winter HOA mass spectra changes over the day and HOA time series patterns exhibit higher diurnal variability than POA (Figs. 4a, 4c, 5). Episodes in winter POA are driven largely by HOA, and suggest that combustion sources other than ubiquitous traffic could be important contributors to HOA.

The winter HOA MS profiles show intra-day differences consistent with period-specific influences. For example, three periods, W171923, W172303, and W170711, show very high *m/z* contributions at *m/z* 60 and *m/z* 73 (>3 times the contributions of other periods) (Table S15). The behaviour of the HOA factor MS in these periods is consistent with the high fractional contributions of biomass burning to the winter POA (exceeds 50% to primary OA) and indicates a burning influence on the HOA MS profile (Table S11). These late-night and early morning periods in Delhi are also likely associated with high frequency of trash burning compared to other hours of the day (Nagpure et al., 2015). We did not separate a trash burning organic aerosol factor (TBOA) likely because TBOA source MS profiles have similar alkyl hydrocarbon fragments as the HOA MS profile (Mohr et al., 2009; Werden et al., 2020), but its influence on MS profiles is apparent. The ratio of contributions of *m/z* 55 to *m/z* 57 is low for all periods (<1.1), which suggests a low influence of cooking. Also, the high contributions at *m/z* 44 in periods between 1500–0300 hours (>2 times the contributions of other periods) lower the ratio of contributions at *m/z* 43 to *m/z* 44 in these periods compared to hours 0300–1500. Interestingly, fractional contributions of winter HOA concentrations to total organic concentrations associated with these periods (1500–0300 hours) are higher than the other three periods (Table S11). These observations indicate that a higher relative composition of HOA in total OA is not necessarily associated with higher clarity of signatures in time-of-day PMF HOA MS profiles.

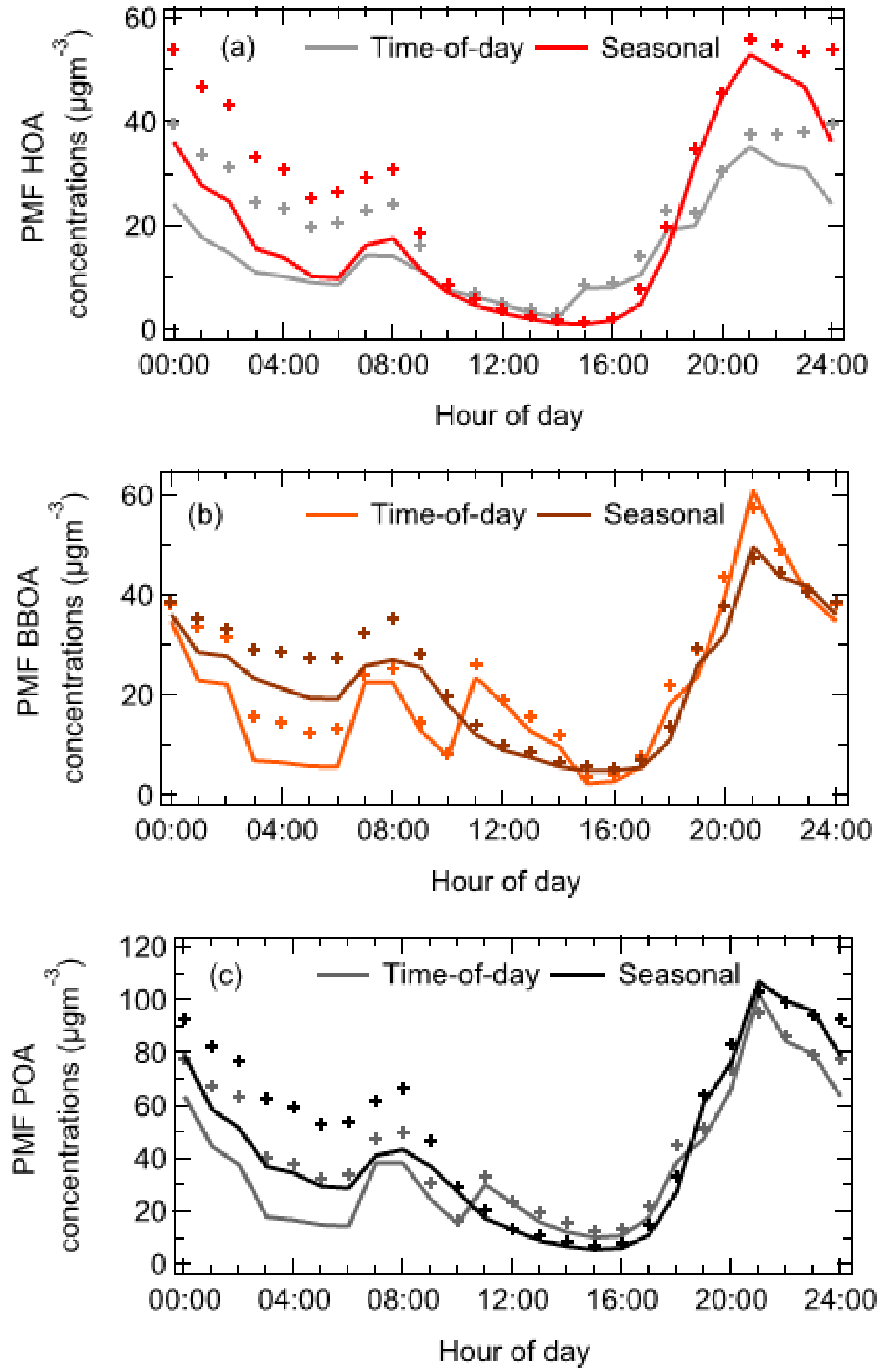

**Figure 4 Seasonally representative hourly averaged diurnal mean (+) and median concentrations (lines) of (a) HOA, (b) BBOA, and (c) POA in winter 2017 from time-of-day and seasonal PMF analysis. HOA exhibits stronger episodes than BBOA.**

Winter HOA time series exhibits strong diurnal variability and late-night–early morning episodes. Traffic peaks occur in the early morning and late at night, corresponding to periods of higher traffic on major traffic corridors, and peaks in HOA concentrations follow that trend (Mishra et al., 2019; Fig. 4). These results are consistent with observations in literature of online submicron aerosol source apportionment conducted in Delhi (Tobler et al., 2020). In contrast with the peak to minimum

ratio of ~10 for POA concentrations, winter peak HOA diurnal concentrations are ~14 times the diurnal minimum and occur at night-time (Fig. 4a, c). This diurnal minimum occurring at midday corresponds to the highest ventilation coefficients experienced in the season (Fig. S1). The large daytime and night-time differences could be attributed to temperature inversions and minimal photochemical conversion of POA to SOA in the evening and at night (Bhandari et al., 2020). The evening (1700–2200 hours) increases and late-night decreases (2200–0600 hours) in winter HOA concentrations are likely associated with the corresponding increases and decreases in traffic congestion on major traffic corridors (Mishra et al., 2019; Nair et al., 2019). Finally, based on the difference between the mean and median, winter HOA exhibits episodic behaviour only at night-time and early morning (2200–0900 hours) and accounts for most of the episodic behaviour first reported in winter POA (Bhandari et al., 2020; Fig. 4a, c). These episodic events could be driven by emissions from polluting heavy commercial vehicles (HCVs) and multi-axle vehicles (MAVs) such as trucks. The counts for these vehicle types peak in these periods (2200–0900 hours) on major traffic corridors, and their emissions follow skewed distributions (Dallmann et al., 2014; Mishra et al., 2019). Previously, it was hypothesized that apart from ubiquitous temporally varying sources such as traffic, episodes in POA could be driven by sources such as burning events (Bhandari et al., 2020). Here, we observe that the episodes are almost exclusively associated with the HOA factor—a combustion factor. We believe that other than traffic, important contributors of combustion exhaust have not been accounted for. Similarity of TBOA MS with HOA MS suggests trash burning could be a contributor (Mohr et al., 2009; Werden et al., 2020). Given their night-time origin (rules out most industrial sources) and low electricity consumption at night (rules out residential diesel generators and power plants), other associated sources could be brick kilns, and construction and road paving activities (Guttikunda and Calori, 2013; Misra et al., 2020; Khare et al., 2021). Future work could identify contributions of different types of source-specific HOA factors to this episodic variability in HOA. Finally, precipitation data exhibits a similar seasonality in diurnal episodes as HOA and could be causing this variability (Fig. S1).

BBOA MS and TS

Here, we find that: (i) the winter BBOA mass spectra show changing MS signatures over the course of the day (Fig. 6), (ii) BBOA time series patterns exhibit higher diurnal variability than HOA and show high concentrations at winter daytime (Figs. 3 and 4b), and (iii) winter BBOA exhibits early morning episodes with lower intensity compared to HOA (Fig. 4a–b).

The winter BBOA MS profiles show intra-day differences consistent with period-specific influences. In all periods in winter, MS of BBOA correlates strongly with the reference BBOA MS profile (Pearson R>0.8; Fig. S5). We observe lowest contributions to winter BBOA MS at *m/z* 60 and *m/z* 73, and the lowest ratio of contributions at *m/z* 43 to *m/z* 44 midday (W171115, ~1.6), pointing to low primary nature of BBOA midday (Table S16). This period also overlaps with periods of high shortwave radiative flux (SWR) and therefore high photochemical processing in the atmosphere (Fig. S1). Midday BBOA MS shows particularly lower contributions at *m/z* 57, suggesting a low influence of traffic. In contrast, we observe the highest ratio of contributions at *m/z* 43 to *m/z* 44 in morning hours (0300–1100 hours), suggesting a very strong primary nature (Table S16); the early morning periods are accompanied by the lowest fractional presence of primary organics (Table S11). With respect to *m/z* 29, BBOA MS exhibits the highest contributions in the period W170307, suggesting a stronger influence of

biomass burning. Low ratios of contributions at *m/z* 55 to *m/z* 57 (<1.3) and *m/z* 41 to *m/z* 43 (<0.8) across all periods suggest limited influence of cooking.

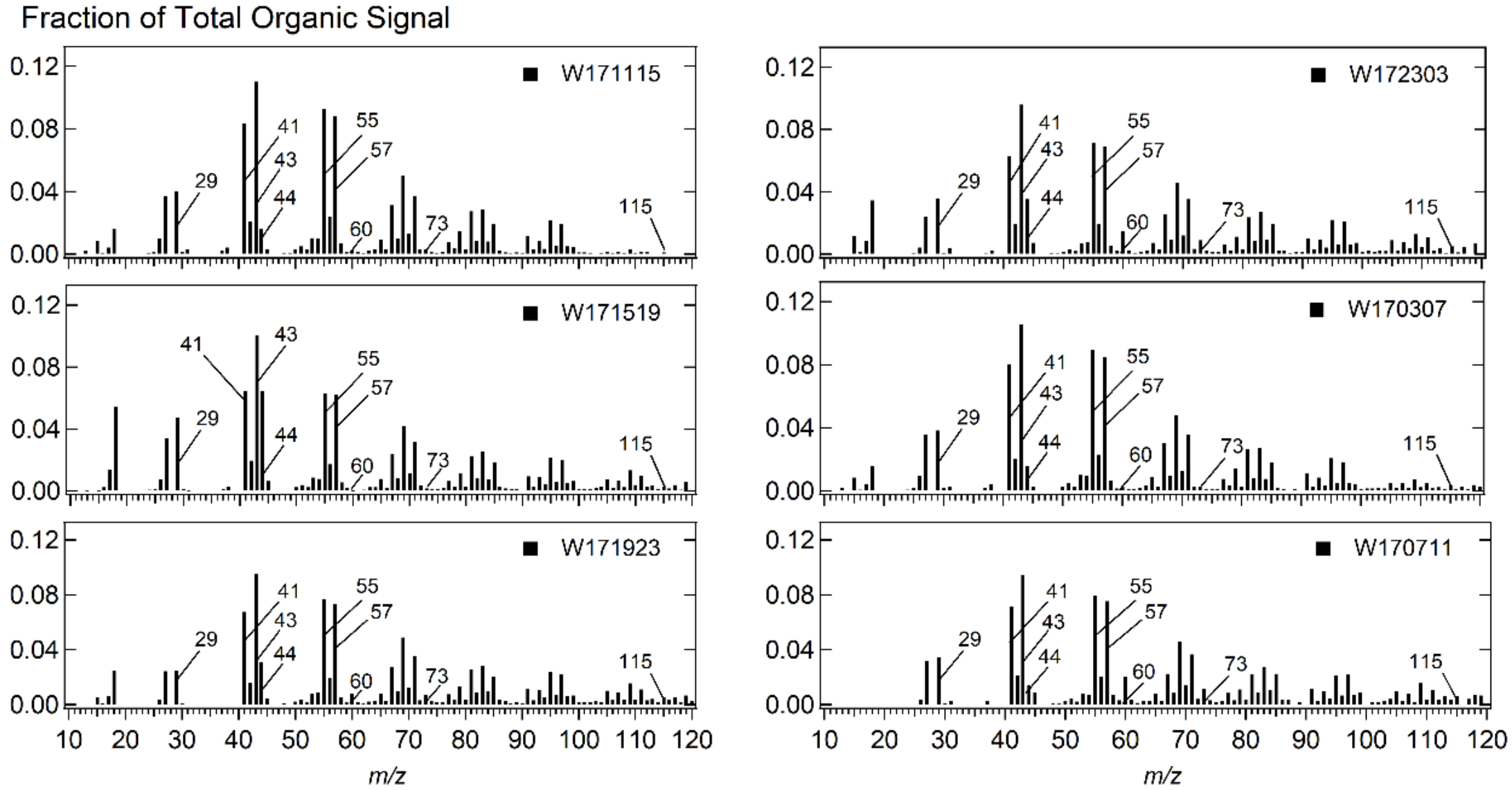

**Figure 5 The mass spectrum of time-of-day PMF hydrocarbon-like organic aerosol (HOA) factor in winter 2017. The mass spectra remain fairly consistent across the time-of-day periods.**

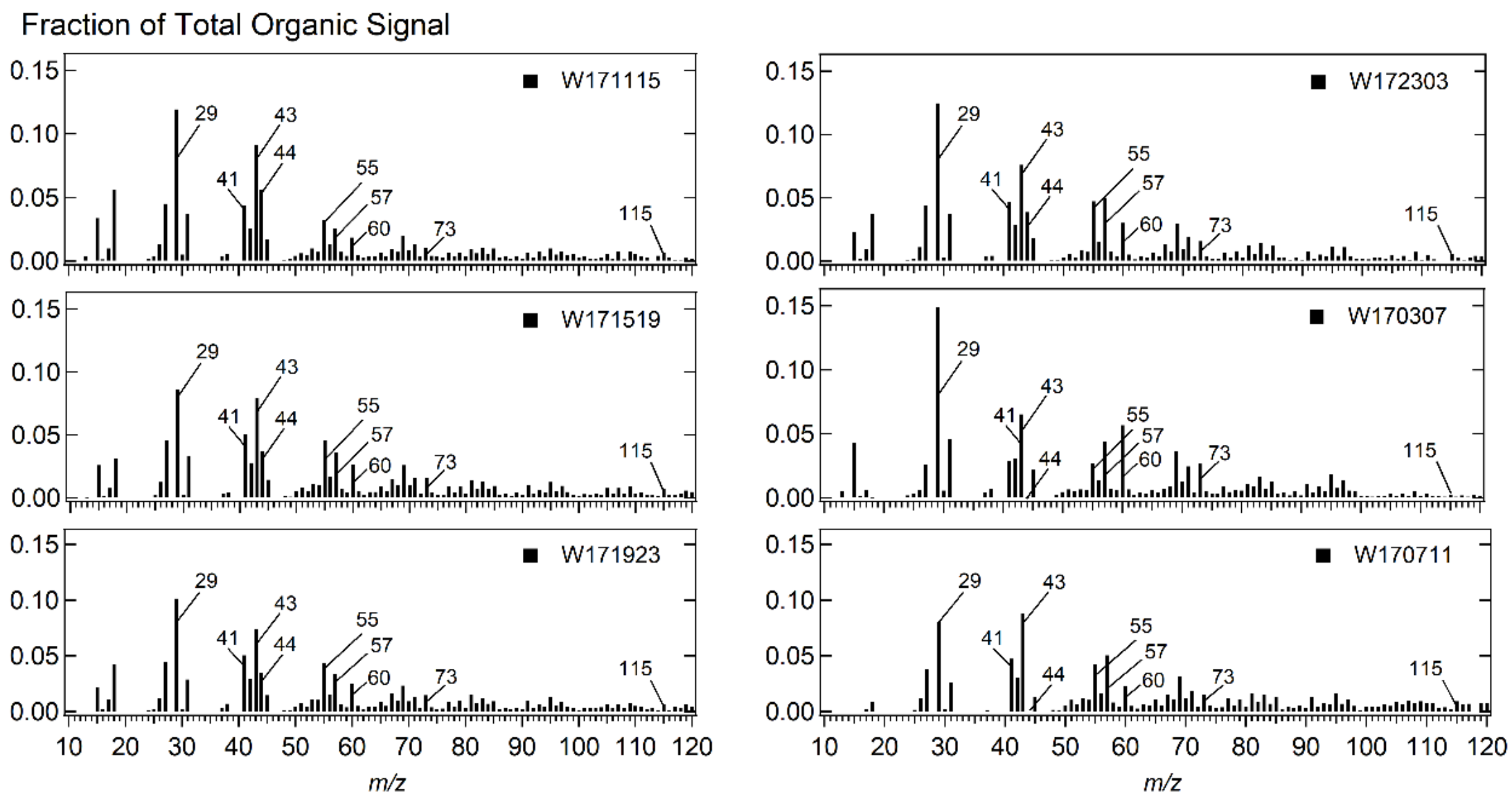

**Figure 6 The mass spectrum of time-of-day PMF biomass burning organic aerosol (BBOA) factor in winter 2017. The mass spectra show important differences across the time-of-day periods.**

Winter BBOA time series exhibits strongly diurnal behaviour, with peaks corresponding to different emission sources. TS of winter BBOA exhibits three peaks—in the early morning (0700–0900 hours), at midday (1100–1200 hours), and at night

(2100–2200 hours) (Fig. 4b). The midday peak in winter BBOA TS suggests that increases in ventilation midday seem to have lesser impacts on trends in biomass burning concentrations compared to traffic, suggesting presence of strong and persistent emissions. Recent work reports early morning and night-time peak of BBOA concentrations in colder seasons (Bhandari et al., 2020); these peaks are likely associated with space heating. Previous studies have not reported the midday peak in winter BBOA concentrations. In this work, the midday peak in BBOA concentrations is associated with the SFC-OA factor; if contributions of BBOA TS to the SFC-OA concentrations (~13 ug/m3) are not considered, this midday period will have the lowest average BBOA concentrations (~5 ug/m3) among all time-of-day periods (Table 1). While similar SFC-OA factors have been reported recently, they have low contributions at midday (Tobler et al., 2020). These differences are likely a limitation of the use of seasonal PMF approaches in these studies (Bhandari et al., 2022). The night-time BBOA peak in concentrations coincides with decreasing ventilation, suggesting its association with inversions (Fig. S1). Winter peak BBOA diurnal concentrations occur at night-time and are ~27 times the diurnal minimum (1500–1600 hours, Fig. 4b), stronger than the diurnal variability of HOA concentrations. The stronger diurnal variability of BBOA than HOA in winter is likely a function of larger emission variability of BBOA sources. Winter BBOA concentrations are higher than HOA in the morning, midday, and particularly at night-time (Fig. 4a, b). These high BBOA concentrations and recent evidence of dark BBOA aging make BBOA an important OOA precursor in Delhi, especially at night-time (Kodros et al., 2020). Daytime temperatures in winter are quite low (Fig. S1), and the volatility of BBOA has been reported on previously (Cappa and Jimenez, 2010; Paciga et al., 2016; Louvaris et al., 2017; Kostenidou et al., 2018). Additionally, biomass burning is frequently used for cooking in Delhi specifically and in India in general (Pant et al., 2015; Rooney et al., 2019; Tobler et al., 2020). Thus, the large winter BBOA time series contributions at daytime are consistent with low temperatures in winter occurring at daytime and the use of biomass burning for cooking (Figs. 3, 4b).

Finally, based on the difference between the mean and median, winter BBOA time series exhibits episodic behaviour only in the early morning hours (0100–0700 hours) (Fig. 4b). The occurrence of these pollution episodes in winter BBOA could be a consequence of precipitation and temperature-related biomass and trash burning (Fig. S1; Nagpure et al., 2015; Werden et al., 2020). Delhi has multiple open waste burning/landfill sites and measurements at these sites show high levoglucosan concentrations, typically considered as tracer for biomass burning (Kumar et al., 2015; Agrawal et al., 2020). Levoglucosan showed similar contributions to total identified sugars at an urban and landfill site, suggesting that BBOA detected could be coming from landfill emissions (Agrawal et al., 2020). Chloride concentrations are associated with landfill emissions and biomass burning as well (Kumar et al., 2015). However, in previous work, we showed that chloride detected at our site is inorganic and has minimal correlation with biomass burning tracers (Bhandari et al., 2020). Here, we observe weak correlations of chloride with winter BBOA concentrations (Pearson R: 0.41, Fig. S8). These results suggest that landfill and trash burning likely have limited contributions at our site. Overall, given the strong diurnal variability of winter BBOA, atmospheric models currently assuming no diurnal variability in emissions likely misrepresent biomass burning (Crippa et al., 2020).

### 3.2.2 Monsoon 2017

In monsoon 2017, we separated HOA or HOA-like factors, and COA or COA-like factors in all time-of-day periods (four separations based on PMF, two separations based on hybrid MLR-PMF) but did not separate biomass burning organic aerosol above detection limits (Table 2; Fig. 7a). Here, we discuss the MS and TS patterns of separated HOA and COA factors in time-of-day periods of monsoon 2017.

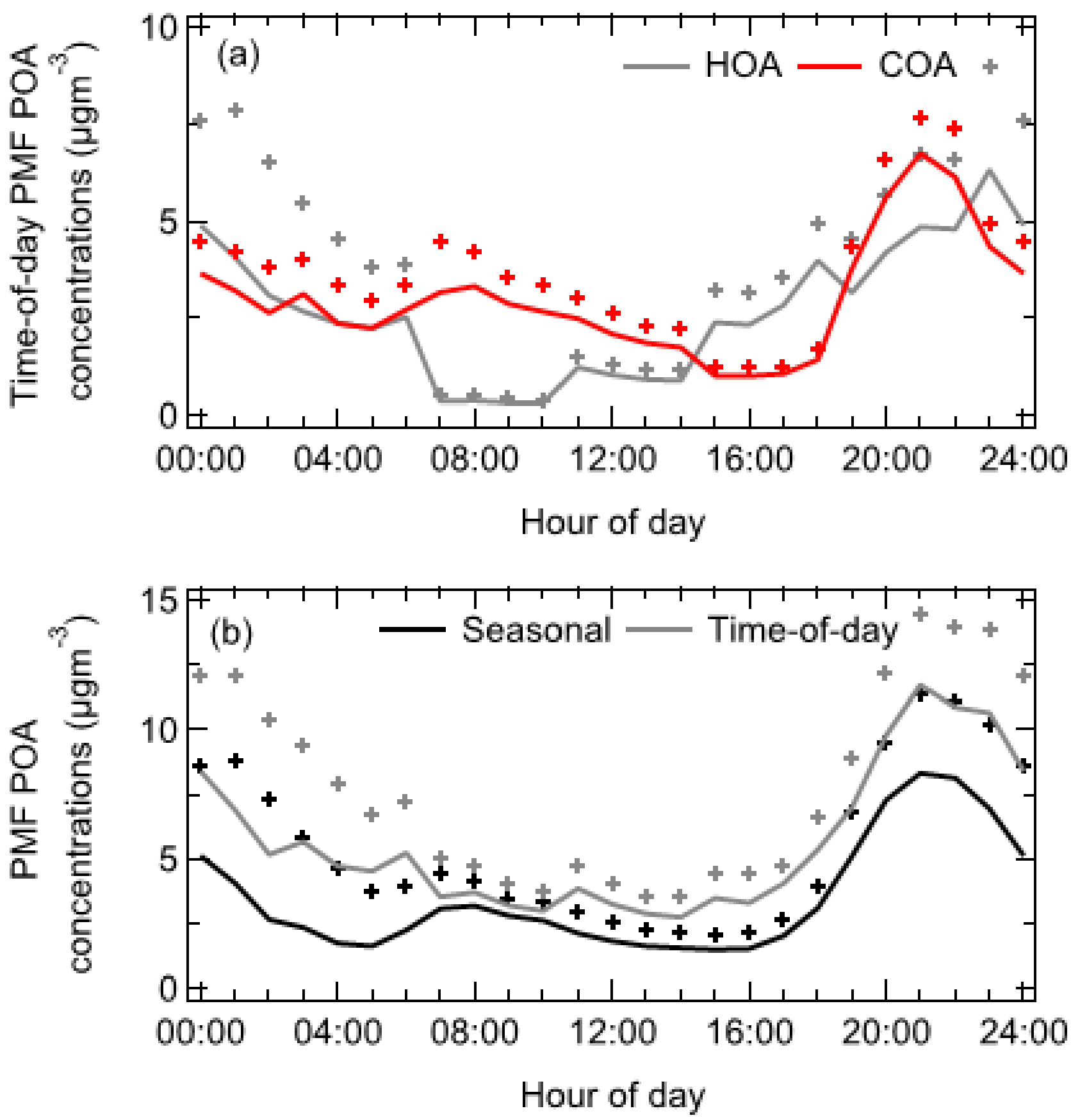

**Figure 7 Seasonally representative hourly averaged diurnal mean (+) and median concentrations (lines) of (a) HOA and COA, and (b) POA in monsoon 2017 from time-of-day and seasonal PMF analysis. HOA exhibits stronger episodes than COA.**

<u>HOA MS and TS</u>

Our results show that the monsoon HOA mass spectra changes over the day and HOA time series patterns exhibit higher diurnal variability than monsoon POA (Figs. 2, 7a, 8). We also show that monsoon HOA concentrations associated with early morning hours and cooking periods in the middle of the day have previously been overestimated and underestimated, respectively. These results are consistent with observations in literature on online submicron aerosol source apportionment conducted at other locations in Delhi in monsoon. HOA also exhibits strong late-night–early morning episodes.

Monsoon HOA MS exhibits variations over the day, with particularly different MS at midday. In all periods in monsoon, MS of HOA correlates strongly with the reference HOA MS profile (Pearson R>0.95, Fig. S4). The MS profiles show highest mass spectral contributions at *m/z* 57 for the period M171923, suggesting a higher influence of traffic (Table S17). The same period also shows the lowest contributions to monsoon HOA MS at *m/z* 41 and the lowest ratio of contributions at *m/z* 55 to *m/z* 57 suggesting the lowest influence of cooking. The ratio of contributions at *m/z* 43 to *m/z* 44 are the lowest in monsoon HOA MS at morning and midday hours (0700–1500 hours), suggesting these periods are detecting aerosols undergoing higher photochemical processing.

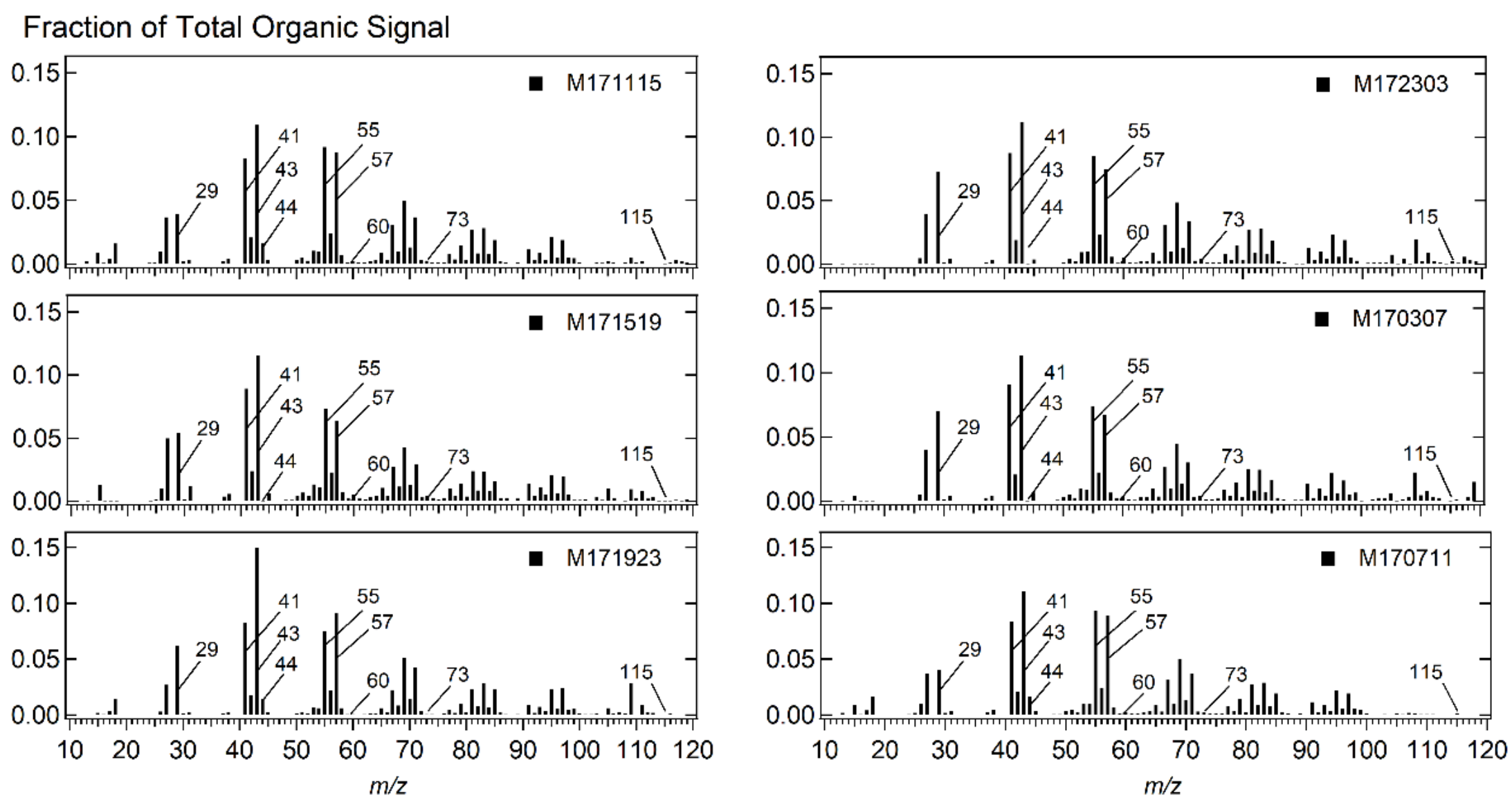

**Figure 8 The mass spectrum of time-of-day PMF hydrocarbon-like organic aerosol (HOA) factor in monsoon 2017. The mass spectra remain fairly consistent across the time-of-day periods.**

Monsoon HOA time series exhibits strong diurnal patterns and episodic behaviour at night-time (Fig. 7a). Monsoon peak HOA diurnal concentrations occur at night-time and are ~20 times the monsoon diurnal minimum. Surprisingly, the diurnal minimum occurs in the early morning traffic hours, and concentrations increase steadily towards night-time. In comparison, peak-to-minimum ratio of monsoon POA concentrations is ~4 (Fig. 7b). After 6 am in monsoon, ventilation coefficient and SWR flux increase dramatically and monsoon HOA decreases (Fig. S1). The matching of the ‘change-points’ in ventilation, SWR flux, and monsoon HOA concentrations points to the strong influence of ventilation (dilution) and SWR flux (reactivity) (UK DEFRA, 2020). These results suggest that the effect of ventilation on traffic-related concentrations is even stronger than previously suggested, especially in monsoon (Bhandari et al., 2020). M170715 (the period from M17 0700–1500 hours) corresponds to the lowest HOA concentration and composition among all time-of-day periods (Tables 2 and S12). Interestingly, monsoon HOA is higher in the middle of the day (1100–1500 hours) than the early morning traffic hours (0700–1100 hours), even though ventilation continues to be high. Similar results indicating a flatter HOA in the morning and presence of a daytime peak in the afternoon have been seen elsewhere as well (Tobler et al., 2020). These results are also consistent

with traffic counts peaking for different modes of traffic at different times during daytime, and daytime traffic congestion peak occurring between 1100–1500 hours seen elsewhere (Mishra et al., 2019; Nair et al., 2019). The HOA increase in evening (1500–2100 hours) is associated with increasing traffic emissions and is followed by a further night-time increase (2100–2400 hours). This further increase is consistent with the admission of heavy-duty vehicles in Delhi permitted only after 9 pm—likely causing the step change in HOA concentrations (The Indian Express, 2017). This pattern is unlike that of POA—concentrations of POA decrease monotonically at night (2100–0300 hours; Fig. 7b), likely because of decreasing emissions (Bhandari et al., 2020). Given that the restrictions are on traffic only, other POA sources such as cooking are not expected to experience this temporary increase and may instead decrease (Table 2, Fig. 7a). Finally, HOA exhibits strong episodic events, especially late at night and in the early morning (1900–0700 hours), and accounts for the episodes previously reported in POA (Fig. 7b; Bhandari et al., 2020). These results are similar to observations in winter HOA TS patterns (Sect. 3.2.1). These episodes could be associated with garbage burning, HCV and MAV traffic, brick kilns, and construction and road paving activities (Guttikunda and Calori, 2013; Nagpure et al., 2015; Mishra et al., 2019; Khare et al., 2021). The relative strength (mean-to-median ratio) of these episodes is higher in monsoon than winter (Figs. 4, 7). Precipitation data exhibits a similar seasonality as the diurnal episodic behaviour of HOA with increasing strength of the diurnal episodes in monsoon. Thus, precipitation could be causing this variability (Fig. S1).

COA MS and TS

The COA mass spectra captures the changing influence of cooking emissions over the day (Fig. 9). COA time series patterns exhibit surprisingly relatively low diurnal variability and low episodic time series behaviour, pointing to its ubiquitous presence in the season (Figs. 2 and 7a). These results are consistent with observations in literature on online submicron aerosol source apportionment conducted at another location in Delhi in warmer months.

COA MS displays clear cooking signatures across the day, and the midday COA MS profile is highly oxidized (Fig. 9). In all periods in monsoon, MS of COA correlates strongly with the reference COA (Pearson R>0.7, Fig. S6). All monsoon COA MS profiles except those corresponding to the morning traffic period (M170711) exhibit a high ratio of contributions at *m/z* 41 to *m/z* 43 pointing to the influence of heated cooking oils (Table S18; Allan et al., 2010; Liu et al., 2018; Zheng et al., 2020). COA MS profiles in all periods show similarly high contributions at *m/z* 60 and *m/z* 73, suggestive of the influence of biomass burning on the cooking MS profile. However, as shown in Sect. 3.1, application of the hybrid MLR-PMF approach did not separate biomass burning above detection limits, likely due to the volatility of BBOA. While there are subtle differences between MS profiles across time-of-day periods, the midday MS profile particularly stands out. The midday monsoon COA MS profile show the lowest mass spectral contributions at *m/z* 57 and the lowest ratio of contributions at *m/z* 43 to *m/z* 44 at midday (0.34), suggesting that the lower primary nature overlaps with periods of high reactivity of the atmosphere (Fig. S1, Table S18). The observations of a POA factor exhibiting oxidized aerosol behaviour (*m/z* 43 << *m/z* 44), particularly in the middle of the day, are indicative of the rapid photochemical processing in Delhi. Based on the observation of highly oxidized local OOA MS profiles, similar conclusions on photochemical processing have been drawn previously as well (Bhandari et

al., 2020). The midday COA MS profile also shows very high contributions at *m/z* 29, suggesting the influence of wood burning associated with cooking.

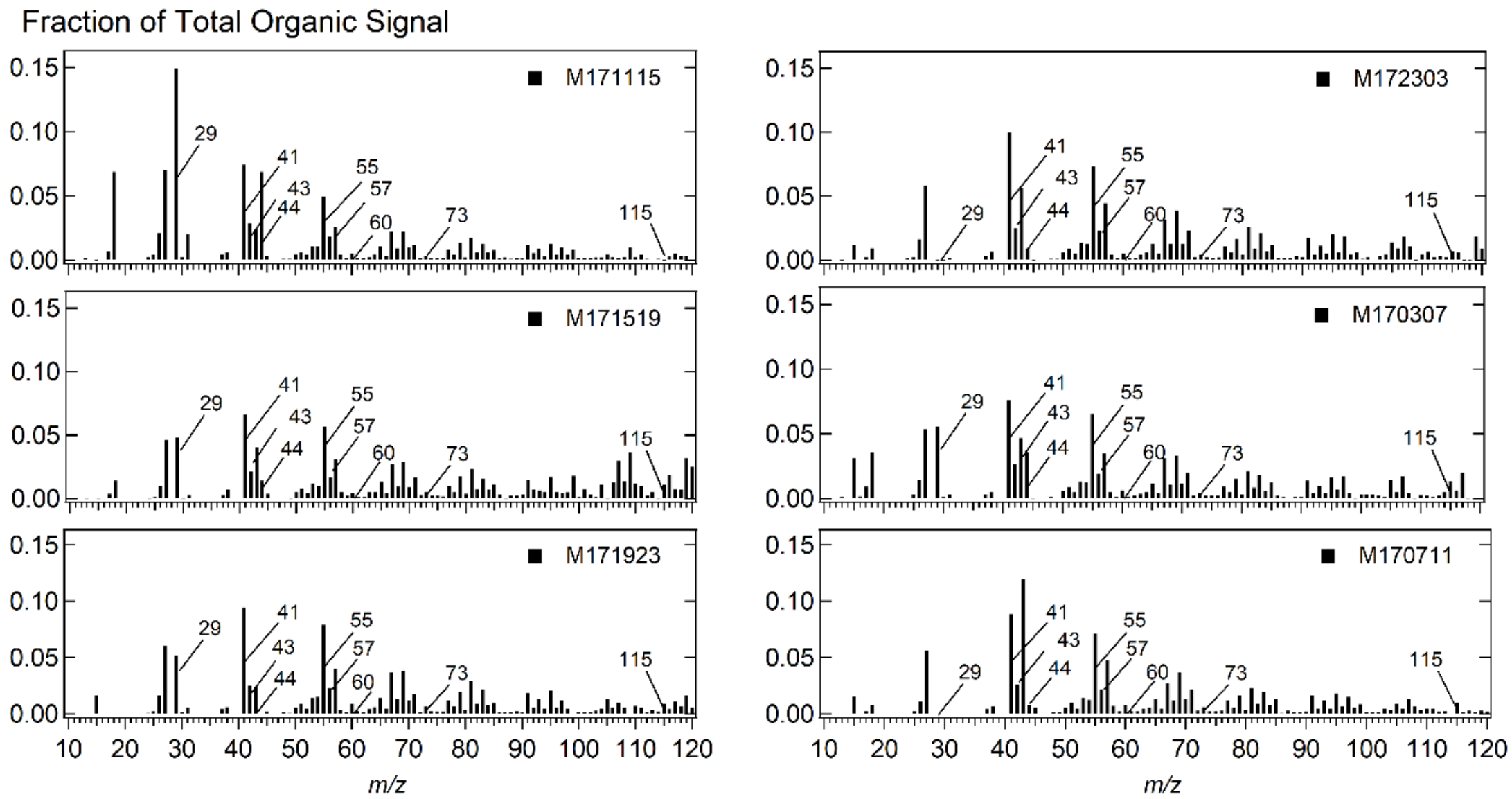

**Figure 9 The mass spectrum of time-of-day PMF cooking organic aerosol (COA) factor in monsoon 2017. The mass spectra show important differences across the time-of-day periods.**

COA TS shows lower diurnal variability than HOA and minimal episodic behaviour. Monsoon peak COA diurnal concentrations at night-time are ~7 times the monsoon diurnal minimum (1600–1700 hours). Previous assumptions of diurnal patterns of cooking expected detectable concentrations at traditional mealtimes only (Zhang et al., 2011; fuel-use activity patterns in Rooney et al., 2019). In contrast, the COA diurnal pattern in this study is consistent with recent publications and has non-negligible time series contributions even at other times (Tobler et al., 2020). Unlike HOA which exhibits only one night-time peak, COA exhibits two peaks, at expected mealtimes—in the early morning (0700–0800 hours) and at night (2100–2200 hours). We did not observe a daytime peak in monsoon HOA TS (Figs. 3 and 7a). Thus, increases in ventilation at daytime seem to have lesser impacts on trends in cooking concentrations compared to traffic, suggesting presence of strong and persistent emissions (Fig. S1). Based on the comparison of mean and median concentrations, COA exhibits minimal episodic behaviour, consistent with the stable patterns of cooking emissions associated with meal consumption (Fig. 7a). Overall, we believe that the large number of food-joints and roadside restaurants operating even at irregular hours in the neighborhood and across the city lead to stable diurnal patterns and reasonable concentrations of COA at all hours.

Recent studies have identified MS similar to COA MS from landfill sites as well (Dall'Osto et al., 2015). They suggest correlations with chloride as marker for contributions from landfill emissions. Delhi has multiple open waste burning/landfill sites and measurements at these sites show high chloride concentrations (Kumar et al., 2015). Garbage burning has been observed in Delhi and could produce emissions resembling landfill emissions (Nagpure et al., 2015). We observe strong

correlations of chloride with COA concentrations (Pearson R 0.65, Fig. S9), suggesting that landfill and garbage burning could be an important contributor to the COA detected by the ACSM.

### 3.3 Comparison of seasonal and time-of-day PMF

Here, we present detailed comparison of the time-of-day PMF and the seasonal PMF approach covering the 24-hour time windows for two seasons. Total POA and OOA mass is obtained as the sum of the POA factor concentrations (HOA, BBOA, COA) and OOA factor concentrations (local OOA and regional OOA factors) respectively. POA (OOA) mass spectra (MS) for the winter and monsoon seasons were calculated by adding the POA (OOA) factors, weighted by their respective time series contributions. We observe correlated TS patterns of the seasonal PMF and time-of-day PMF POA and OOA but large differences in the MS (Figs. S10–S11a–b, S12–S15a–f). These results suggest that the differences between the two approaches are driven by the ability of the time-of-day PMF approach to capture variable MS profiles over the day.

The seasonal PMF analysis yields three or four factors, depending on the season. In winter, we obtained two POA factors: HOA and BBOA, and two OOA factors, local OOA and regional OOA. In monsoon, we obtained one POA factor: a mixed HOA-COA factor, and two OOA factors, local OOA and regional OOA. In both seasons, mass spectrum of the POA factor from seasonal PMF analysis correlates most strongly with the reference HOA, and/or COA/BBOA ($R$>0.85) (Figs. S16 and S17). OOA MS correlates most strongly with the reference low-volatility oxygenated organic aerosol (LVOOA) MS profile ($R$>0.95) (Figs. S18 and S19). As shown in Figs. S10–11a–b, the behaviour of POA and OOA TS in time-of-day PMF is similar to that of seasonal PMF POA and OOA TS, respectively (W17 POA: slope ~ 0.83, intercept ~ 1.6, $R$~0.97; W17 OOA: slope ~ 1.26, intercept ~ −7.0, $R$ ~ 0.88; M17 POA: slope ~ 1.15, intercept ~ 1.5, $R$~0.97; M17 OOA: slope ~ 0.91, intercept ~ −0.5, $R$ ~ 0.98). MS of POA and OOA in time-of-day PMF is broadly similar to seasonal POA and OOA MS due to the design of factor identification (MS profiles should be correlated to reference MS profiles, Sect. S1 in Bhandari et al., 2022). However, we observe substantially different contributions at multiple *m/zs* in both seasons (Figs. S12–S15a–f). We discuss the similarities and differences of MS and TS patterns in more detail below.

#### 3.3.1 Comparisons of POA MS and TS obtained using time-of-day PMF and seasonal PMF

The time-of-day PMF POA shares similarities with seasonal PMF POA (Table S13, Figs. S10–S11. S12a–f, S14a–f). However, specific periods are marked by differences that are characteristic of sources, meteorology and/or reaction chemistry corresponding to that period.

<u>Winter POA MS and TS</u>

The comparison of winter 2017 POA using the time-of-day PMF and seasonal PMF approaches shows several similarities, time-of-day independent differences, as well as time-of-day dependent differences in MS contributions at key *m/zs* and TS patterns.

Time-of-day PMF winter POA MS shows a stronger primary nature, lower influence of biomass burning, and a stronger influence of cooking compared to seasonal PMF winter POA MS. We also show that seasonal PMF analysis

overestimates BBOA concentrations associated with early morning periods and underestimates night-time HOA concentrations (Tables 1 and S13). We observe similar or slightly higher *m/z* 41 in POA MS in all periods in winter time-of-day PMF compared to seasonal PMF, suggestive of a stronger influence of cooking (Tables S19–S20). However, we did not separate cooking as a factor in any period in winter, likely due to low concentrations (Sect. 3.1). Winter time-of-day PMF analysis results in lower or comparable contributions of *m/zs* 29, 60, and 73 to POA MS compared to the seasonal PMF analysis (lower by as much as 40%) (Tables S19 and S20). These results suggest a lower influence of wood and biomass burning; also, time-of-day PMF apportions higher *m/z* 29 to secondary organics (Tables S23 and S24). Overall, we observe lower or similar average BBOA concentrations in all time-of-day periods of winter 2017 (except midday) in time-of-day PMF compared to seasonal PMF, indicating a lower influence of biomass burning (Tables 1 and S13). The contrasting behaviour of winter midday BBOA is likely due to high BBOA contributions from SFC-OA. The early morning periods (0300–1100 hours) show time-of-day PMF BBOA concentrations lower than seasonal PMF BBOA concentrations by ≥50%.

We observe several time-of-day dependent differences in MS as well. Early morning time-of-day PMF POA MS profiles of W170307 and W170711 show a higher ratio of contributions at *m/z* 43 to *m/z* 44 (>10 compared to ∼3.3) and contributions at *m/z* 55 and *m/z* 57 higher by ∼10–20%, indicating a stronger primary nature and clearer signatures in POA MS in winter time-of-day PMF compared to seasonal PMF (Tables S19 and S20). Similar to the patterns of HOA concentrations for the periods of W170307 and W170711, the fractional contribution of POA concentrations to total OA concentrations is the lowest among all time-of-day periods (Table S11). Thus, a higher relative strength of POA concentrations in total OA concentrations is not necessarily associated with higher clarity of signatures in time-of-day PMF POA MS profiles relative to the seasonal PMF POA MS (discussed for HOA in Sect. 3.2.1). These early morning periods are at the transition of a decreasing inversion and show POA concentrations in time-of-day PMF analysis that are lower by ≥30% than seasonal PMF analysis (Table S13, Fig. S1). Winter periods with cooking influence (1100–2300 hours) display POA MS profiles with larger ratio of contributions of *m/z* 55 to *m/z* 57 in comparison to seasonal PMF POA MS profiles, in line with the strong cooking influence expected in these periods (Tables S19 and S20). The afternoon and evening time-of-day PMF POA MS profiles (W171115 and W171519) show lower MS contributions at *m/zs* 29, 55, 57, and 60, and higher MS contributions at *m/z* 43 and *m/z* 44, with a lower contribution ratio of *m/z* 43 to *m/z* 44 (<2) compared to seasonal PMF (>3). Together with the higher ratio of contributions at *m/z* 55 to *m/z* 57, these results suggest a lower influence of traffic, wood and biomass burning, but a stronger influence of cooking and oxidization processes in the atmosphere. These results are in line with the similarity of the time-of-day PMF POA MS profile for W171115 with an SFC-OA MS profile reported recently (Bhandari et al., 2022). These midday periods have a high ventilation coefficient and have primary concentrations higher in time-of-day PMF by ≥40%, the largest relative difference in POA concentrations between seasonal PMF analysis and time-of-day PMF analysis among all time-of-day periods (Fig. S1, Table S13). Because of the low total OA concentrations in these periods, they likely have limited importance in seasonal PMF analysis with respect to determining the overall seasonal mass spectra and time series patterns. The late-night primary MS profiles of W171923 and W172303 show the strongest similarities with the seasonal PMF MS profile among all time-of-day periods (Tables S19 and S20, Figs. S12a–f). These late-night periods experience increasing

inversion and show primary concentrations within 15% of seasonal analysis (Fig. S1, Table S13). Interestingly, night-time HOA concentrations (1900–0300 hours) are lower by ~30% relative to seasonal PMF HOA (Tables 1 and S13). Thus, even in periods with relatively similar POA concentrations and POA MS in time-of-day PMF and seasonal PMF, the apportionment to HOA and BBOA is substantially different.

Monsoon POA MS and TS

In monsoon 2017, the comparison of POA MS and TS using the time-of-day PMF and seasonal PMF approaches shows several consistent differences across the day as well as time-of-day dependent differences in MS contributions.

Monsoon time-of-day POA MS shows a similarly strong primary nature but a lower influence of biomass burning, and a stronger influence of cooking compared to seasonal PMF analysis. Monsoon time-of-day PMF analysis results in very high ratio (~6–31) of *m/z* 43 to *m/z* 44 contributions in POA MS in all periods except midday (M171115), consistent with the observed zero *m/z* 44 in the seasonal PMF MS profile (Tables S21 and S22). At the same time, *m/z* 43 is lower in POA MS from the time-of-day PMF analysis compared to seasonal PMF analysis in all but one time-of-day period of M170711. Additionally, *m/z* 57 is lower in all six time-of-day periods compared to seasonal PMF, suggesting a lower influence of traffic and combustion exhaust. We also observe a higher POA concentration (higher by ~20–80%) in time-of-day PMF compared to seasonal PMF in all 6 periods, with larger relative difference observed in periods with the lower OA concentrations (M170307, M171115, M171519) (Table S14). This observation together with the lower influence of HOA concentrations in POA suggests enhanced importance of other primary emissions. The cooking tracer *m/z* 41 in time-of-day PMF POA MS is lower than seasonal PMF POA (Tables S21 and S22). Interestingly, the ratio of contributions at *m/z* 41 to *m/z* 43 is higher in monsoon time-of-day PMF POA MS compared to seasonal PMF POA MS. In the same vein, the ratio of contributions at *m/z* 55 to *m/z* 57, typically considered as a tracer for the influence of cooking emissions is higher in monsoon time-of-day PMF POA MS than seasonal PMF. These results suggest a stronger influence of cooking in time-of-day PMF analysis. This finding is consistent with the separation of COA in all six time-of-day periods of monsoon 2017, unlike the seasonal PMF run, which had a POA (mixed HOA-COA) factor (Table S3).

We show that seasonal PMF analysis underestimates POA concentrations throughout the day and midday seasonal POA concentrations seem to miss HOA concentrations (Tables 2 and S14, Fig. 3). Unlike winter, time-of-day periods in monsoon have very few time-dependent differences in time-of-day MS profiles relative to the seasonal results. Based on *m/z* 29 contributions in two periods (M170307 and M171115), we observe a stronger influence of biomass burning (Tables S21 and S22). The night-time POA MS profiles of M171923 and M172303 show the strongest similarity with the seasonal POA MS profiles (Figs. S14a–f). Finally, the diurnal TS patterns for monsoon seasonal PMF POA at midday (1100–1900 hours) match the monsoon time-of-day PMF COA perfectly, and time-of-day PMF also suggests presence of additional HOA in this period (Fig. 2). This observation suggests that traffic emissions associated with cooking periods (e.g., cooking-associated combustion exhaust) might have been underestimated in seasonal PMF analyses (Bhandari et al., 2020).

### 3.3.2 Comparisons of OOA MS and TS obtained using time-of-day PMF and seasonal PMF

In time-of-day PMF conducted on winter and monsoon, we separated two–three OOA factors in all periods (Table S3). Recent results suggest that these separated OOA factors do not represent separate OA sources (Drosatou et al., 2019). Here, we only discuss total OOA MS and TS from time-of-day PMF and seasonal PMF. Total OOA mass is obtained as the sum of the OOA factors. OOA MS for the winter and monsoon seasons were calculated by adding the OOA factors, weighted by their respective time series contributions.

Similar to characterization of primary factors, MS can be used to characterize OOA as well. A large body of evidence that has characterized aging of HOA and BBOA suggests that these primary SOA precursors produce similar OOA factor MS (Sage et al., 2008; Grieshop et al., 2009; Zhang et al., 2011; Kroll et al., 2012; Platt et al., 2013; Platt et al., 2014; Liu et al., 2019). However, a distinctive feature of OOA from BBOA precursors is a strong presence at *m/z* 60, which can be used to identify oxidized BBOA presence (Weimer et al., 2008; Grieshop et al., 2009; Ahern et al., 2019). In recent years, similar aging experiments have been conducted for cooking organic aerosols as well. While rapid aging is observed for COA precursors such as heated cooking oils, aging is not accompanied with high oxidation states of OOA (Kaltsonoudis et al., 2017; Liu et al., 2017a; Liu et al., 2017b; Liu et al., 2018; Zhang et al., 2021). Additionally, for oxidized COA MS, high contributions at *m/z* 41 are a distinguishing feature relative to oxidized HOA MS and BBOA MS. In Delhi, we observe highly oxidized OOA; all OOA MS profiles detected in Delhi are highly correlated to the reference LVOOA MS profile (Pearson R>0.95, Figs. S16–19). Here, we present a comparison of time-of-day PMF OOA MS and TS to the seasonal PMF OOA MS and TS.

<u>Winter OOA MS and TS</u>

Here, we discuss the MS and TS patterns of total OOA in time-of-day periods of winter 2017 and compare them to seasonal PMF results (Figs. S13a–f, Tables S13, S23–S24). Our results indicate that (i) the highest oxidation state of winter time-of-day PMF OOA MS, based on the ratio of contributions at *m/z* 43 to *m/z* 44, occurs in the afternoon and lowest occurs at night-time (Tables S23 and S24), (ii) diurnal changes occurring in the winter OOA mass spectra at *m/z* 44 in time-of-day PMF relative to the seasonal PMF approach can be explained by photochemical activity, and (iii) winter time-of-day PMF OOA time series patterns exhibit significantly lower diurnal variability than time-of-day PMF POA but stronger diurnal variability than seasonal PMF OOA (Fig. 10a).

Winter OOA MS exhibits variations over the day, with particularly different MS midday. In all periods in winter, MS of OOA factors correlates strongly with the reference OOA MS profile (Pearson R>0.95, Figs. S16 and S18). The MS profiles show low mass spectral contributions at *m/z*s 57, 60, and 73 midday (1100–1900 hours), suggesting low primary nature of the OOA MS profile (Table S23). In the winter OOA MS profiles, contributions at cooking tracer *m/z* 41 are low, suggesting that cooking has minimal remaining influence on the oxidized aerosols. We observe smooth variations in the metric of oxidation state of aerosols—the ratio of contributions at *m/z* 43 to *m/z* 44 in the winter OOA MS profiles. The ratio is lowest midday (1100–1900 hours) and is highest at night-time (1900–0300 hours). The low midday (high night-time) ratios are primarily

driven by high (low) contributions at *m/z* 44. The lowest values of the ratio overlap with periods of high atmospheric processing, based on the SWR flux (Fig. S1) and the highest values of the ratio overlap with traffic congestion and peak traffic counts for several vehicle types on major traffic corridors (Mishra et al., 2019; Nair et al., 2019). In contrast with the differences in MS in POA factors, OOA MS shows limited variability between the time-of-day PMF and seasonal PMF (Figs. S13a–f, Tables S23 and S24). Midday periods (1100–1900 hours) show higher contributions at *m/z* 44 (+10%) and lower contributions at *m/z* 43 (-10%) in time-of-day PMF analysis-based OOA MS, in line with the rapid photochemical processing at these times (Fig. S1). At the same time, for these midday periods, time-of-day PMF predicts OOA concentrations lower by ~10–20% than seasonal PMF (Table S13). Time-of-day PMF OOA MS of early morning periods (0300–1100 hours) show a lower contribution at *m/z* 44 (-15%), whereas time-of-day PMF OOA TS shows higher OOA concentrations (higher by ~20–30%) (Tables S13, S23, and S24). Finally, the lowest contributions to OOA MS at the key *m/z*, *m/z* 44, are observed at night-time periods (1900–0300 hours, -20%), in line with the low photochemistry at these times. For these periods, time-of-day PMF predicts OOA concentrations higher by ~30% (Table S14). Thus, lower OOA concentrations are seemingly associated with higher clarity of signatures in the comparison of time-of-day PMF OOA MS and seasonal PMF POA MS profiles.

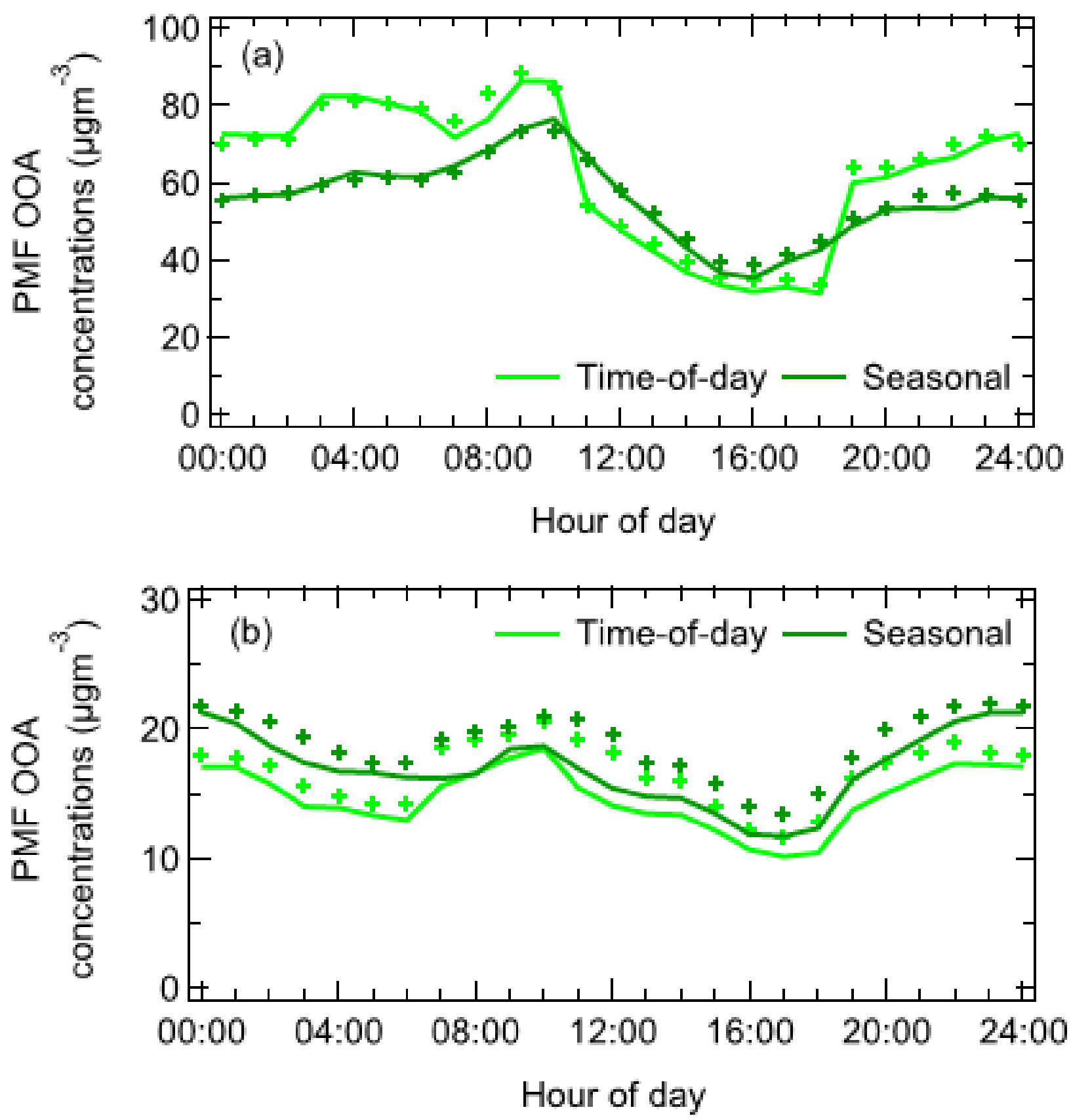

**Figure 10 Seasonally representative hourly averaged diurnal mean (+) and median concentrations (lines) of time-of-day PMF OOA and seasonal PMF OOA in (a) winter 2017 and (b) monsoon 2017. OOA concentrations show limited diurnal variability in both time-of-day PMF and seasonal PMF analyses.**

In other words, higher the OOA concentration detected in time-of-day PMF relative to seasonal PMF, lower its oxidation state. This observation is likely linked to the partitioning of semi-volatile oxidized compounds to the particle phase at higher concentrations of OA. These results are reflected in correlations of time-of-day PMF OOA MS profiles with the reference SVOOA MS profile as well, with the correlations increasing as total OOA concentration increases (Table S13, Fig. S18).

Time-of-day PMF winter OOA TS shows several similarities, time-of-day independent differences, as well as time-of-day dependent differences in TS contributions (Figs. 10a). Time-of-day PMF winter OOA concentrations exhibits limited diurnal patterns, similar to seasonal OOA. The limited diurnal patterns are a result of increasing photochemical processing early in the day, effect of ventilation leading to rapid dampening midday, and high concentrations at night driven by high primary emissions (Bhandari et al., 2020). However, absolute OOA concentrations show a large contrast in the two techniques, especially at midday. For the time-of-day PMF approach, winter peak OOA diurnal concentrations in the morning (0900–1000 hours) are ~2.7 times the diurnal minimum (which occurs in the evening, 1800–1900 hours); substantially greater than the ~2.2 observed for seasonal PMF winter OOA concentrations. This difference is driven by lower OOA concentrations midday (1100–1900 hours) and higher OOA concentrations at other hours.

Monsoon OOA MS and TS

Here, we discuss the MS and TS patterns of total OOA in time-of-day periods of monsoon 2017 and compare them to seasonal PMF results (Figs. S15a–f, Tables S14, S25–S26). Our results indicate high oxidation state of OOA MS (based on the ratio of contributions at *m/z* 43 to *m/z* 44) throughout the day. OOA concentrations show similar diurnal patterns between time-of-day PMF and seasonal PMF and OOA concentrations are always lower in time-of-day PMF.

Monsoon OOA MS is particularly striking midday, likely due to the influence of biogenic emissions. In all periods in monsoon, MS of OOA factors correlates strongly with the reference OOA MS profile (Pearson R>0.95, Figs. S17 and S19). The OOA MS profiles show low mass spectral contributions at cooking tracer *m/z* 41 and biomass burning tracer *m/z* 60, suggesting that cooking and biomass burning has minimal remaining influence on the oxidized aerosols (Table S25). Overall, OOA MS in monsoon time-of-day periods show low contribution at *m/z* 43 and high contribution at *m/z* 44. Similar to winter, we see a continuum of oxidation state, with the ratio of contributions at *m/z* 43 to *m/z* 44 in monsoon OOA MS experiencing peak and lowest values in consecutive periods (1100–1500 hours and 1500–1900 hours) in the middle of the day. The peak value of the *m/z* 43 to *m/z* 44 ratio midday (1100–1500 hours) is driven by a higher contribution at *m/z* 43 (0.08, relative to an average of 0.07 in other periods) and lower contributions at *m/z* 44 (0.15, relative to an average of 0.16 in other periods). The higher values of *m/z* 43 in midday OOA MS are likely caused by biogenic emissions generating semi-volatile compounds—midday OOA MS profile shows the strongest correlation with the reference SVOOA MS among all time-of-day periods (Canonaco et al., 2015; Pearson R~0.8, Fig. S19). OOA MS in the other midday period (1500–1900 hours) has the lowest ratio of contributions at *m/z* 43 to *m/z* 44 and the weakest correlation with the reference SVOOA MS among all time-of-day periods, and OOA MS shows behaviour consistent with high photochemical processing (Fig. S1; Pearson R~0.7, Fig. S41; Table S25). In comparison to the difference in MS between time-of-day PMF and seasonal PMF in monsoon POA factors,

monsoon OOA factor MS show limited variability between time-of-day PMF and seasonal PMF (Tables S25 and S26). Additionally, time-of-day PMF OOA MS in all periods show higher (lower) contributions at *m/z* 44 (*m/z* 43) compared to seasonal PMF OOA MS, pointing to the higher oxidation state of OOA MS in time-of-day PMF. Except midday hours (1100–1500 hours), the oxidation state of OOA MS is also higher than for corresponding periods in winter (ratio of contributions at *m/z* 43 to *m/z* 44 higher by 15%), likely a function of the higher temperatures and photochemical activity in monsoon compared to winter (Tables S23, S25).

Time-of-day PMF monsoon OOA TS shows very limited diurnal variability and largely similar patterns as seasonal PMF (Fig. S10b). Mean diurnal concentrations of monsoon OOA vary between 10 and 20 µg $m^{-3}$, similar to seasonal OOA, and the peak to minimum ratio is within 10% between the two approaches (seasonal: 1.8; time-of-day: 1.7). It is therefore not surprising that absolute OOA concentrations between the two approaches are within 20% throughout the day, with the time-of-day PMF OOA concentrations always lower.

## 4 Conclusions

This study provides source apportionment results for two seasons, winter and monsoon of 2017, for a receptor site in New Delhi, the most polluted megacity in the world. A new technique conducting PMF analyses by resolving data by time-of-day is deployed for the two seasons and results are compared with the seasonal PMF approach typically applied on atmospheric datasets. Time-of-day PMF analyses yields three to five factors—one to two POA factors (two of HOA, BBOA, and COA), and two to three OOA factors. HOA, the fuel combustion and traffic POA surrogate, occurs in every season, shows strong diurnal patterns, with high concentrations early in the morning and late at night. Monsoon HOA does not have a daytime peak, likely due to high temperatures and lower concentrations. BBOA factor, a surrogate for biomass burning, separates only in winter, exhibits three peaks—in the early morning, at midday, and late at night, and is likely associated with space heating and solid-fuel combustion. BBOA diurnal patterns are stronger in variability than HOA diurnal patterns. The COA factor, a surrogate for cooking, separates only in monsoon and reports stable diurnal patterns and significant concentrations throughout the day, suggesting the presence of cooking sources all day. POA episodes in the late-night and early morning hours of the day are driven by HOA and BBOA in winter and HOA in monsoon and suggest the influence of precipitation and presence of stochastic sources. Analysis conducted using the volatility basis set suggests that differences in ventilation coefficient and temperature, and therefore, equilibrium gas-particle partitioning can explain the differences in factor separation between winter and monsoon.

All factors exhibit variable MS over the day, with clear influence of period-specific sources on the MS. For example, winter midday primary MS profiles show signatures of SFC-OA. These results put this work in contrast to the seasonal PMF approach, wherein a single factor MS represents each factor throughout the season. The ability to capture variable MS results in time-dependent changes in the concentrations and composition of primary and secondary PMF factors as well. For example, all periods in monsoon and midday periods in winter predict higher fractions of primary organics in time-of-day PMF compared

to seasonal PMF. Another interesting case is that of the winter time-of-day PMF OOA factors, which show anti-correlation of OOA concentrations with oxidation state, suggesting the influence of semi-volatile oxidized organics.

We observe HOA and BBOA contribute 16% and 19% respectively to total organic non-refractory submicron aerosols ($NR-PM_1$) in winter 2017. In monsoon, average organic $NR-PM_1$ contributions of HOA and COA are 14% and 15%, respectively. These results are in broad agreement with several top-down (receptor modeling) and bottom-up (source-oriented) studies— annual contributions to primary organics from biomass burning and cooking are larger than or comparable to traffic in Delhi. COA and BBOA are expected to be associated with residential cooking and heating emissions. We hope that the separation of cooking and biomass burning as factors in our study allows policymakers to utilize results from this study on source contributions to formulate action plans. The large contributions of COA and BBOA at our receptor site highlight that air quality management in Delhi should tackle the issues of outdoor and indoor air quality simultaneously. A recent federal government ordinance establishing the Commission for Air Quality Management and providing the commission with "exclusive" jurisdiction" for air pollution-related decision-making superseding those of local bodies is an encouraging step in the right direction (Govt. of India Ministry of Law and Justice, 2020). Future work could utilize these highly time-resolved speciated measurements to extract important missing factors in online mass spectrometry source apportionment studies in Delhi such as coal combustion organic aerosol.

## Appendix A: Abbreviations

| | |
|---|---|
| ACSM | Aerosol Chemical Speciation Monitor |
| AMS | Aerosol mass spectrometer |
| BBOA | Biomass-burning organic aerosol |
| BS | Bootstrapping |
| BS-DISP | Bootstrapping enhanced with displacement |
| CMB | Chemical Mass Balance |
| COA | Cooking organic aerosol |
| DAS | Delhi Aerosol Supersite |
| DISP | dQ-controlled displacement of factor elements |
| EC | Elemental carbon |
| EDGAR | Emissions Data for Global Atmospheric Research |
| GBD | Global Burden of Disease |
| HCVs | Heavy commercial vehicles |
| HOA | Hydrocarbon-like organic aerosol |
| IIT | Indian Institute of Technology |

| | |
|---|---|
| LT | Local Time |
| MAPS | Major Air Pollution Sources |
| MAVs | Multi-axle vehicles |
| ME-2 | Multilinear Engine |
| MLR | Multi-linear regression |
| MS | Mass spectral profiles |
| NCR | National Capital Region |
| NR-$PM_1$ | Non-refractory submicron particulate matter |
| NR-$PM_{2.5}$ | Non-refractory PM smaller than 2.5 µm in diameter |
| OC | Organic carbon |
| OM | Organic mass |
| OOA | Oxygenated organic aerosol |
| Org | Organic |
| PET | PMF evaluation tool |
| PAHs | Polycyclic aromatic hydrocarbons |
| PM | Particulate matter |
| $PM_1$ | Submicron particulate matter |
| $PM_{2.5}$ | Particulate matter smaller than 2.5 µm in diameter |
| PMF | Positive matrix factorization |
| POA | Primary organic aerosol |
| POC | Primary organic carbon |
| SFC-OA | Solid fuel combustion organic aerosol |
| SOA | Secondary organic aerosol |
| SVOOA | Semi-volatile oxygenated organic aerosol |
| SWR | Shortwave radiative flux |
| T | Temperature |
| TBOA | Trash burning organic aerosol |
| TERI | The Energy and Resource Institute |
| TS | Time Series |
| UVPM | Ultraviolet-absorbing particulate matter |
| VBS | Volatility basis set |
| VC | Ventilation coefficient |

**Data availability**

Underlying research data are available by request to Lea Hildebrandt Ruiz: lhr@che.utexas.edu and will be posted to the Texas Data Repository upon manuscript acceptance.

**Author contributions**

LHR, JSA, GH, and SB designed the study. SB and ZA carried out the data collection. SB carried out the data processing and analysis. SB, JSA, and LHR assisted with the interpretation of results. All co-authors contributed to writing and reviewing the paper.

**Competing interests**

The authors declare that they have no conflict of interest.

**Acknowledgements**

We are thankful to the Indian Institute of Technology (IIT) Delhi for institutional support. We are grateful to all student and staff members of the Aerosol Research Characterization Laboratory (especially Prashant Soni, Nisar Ali Baig, and Mohammad Yawar) and the Environmental Engineering Laboratory (especially Sanjay Gupta) at IIT Delhi for their constant support. We are thankful to Philip Croteau (Aerodyne Research) for always providing timely technical support for the ACSM, and Dr. Penttti Paatero, Dr. Phil Hopke (University of Rochester), and Dr. Dave Sullivan (UT Austin) for insightful conversations about PMF. I would also like to thank Dr. Nancy Sanchez (now, Chevron; then. at Rice U.) for discussions at the UT Austin Texas Air Quality Symposium that inspired this work. Lastly, I thank Shahzad Gani (University of Helsinki) for leading the instrument setup for the Delhi Aerosol Supersite study.

**Financial support**

This manuscript is based on work supported in part by the Welch Foundation under Grants F-1925-20170325 and F-1925-20200401 and the National Science Foundation under Grant 1653625. Joshua S. Apte was supported by the ClimateWorks Foundation. We thank the funding agencies for their support.

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
