# Peer review of "Contributions of primary sources to submicron organic aerosols in Delhi, India"

_Atmospheric Chemistry and Physics, 2022_

## Author Comment (AC1)

We thank the reviewers for their comments. All comments are addressed below. Reviewers' comments are included in italics, our responses are included in blue, and updated manuscript text is included in red.

**Reviewer 1**

*The study by Bhandari et al. builds on the group's previous and companion work measuring and characterizing temporal patterns and source apportionment of non-refractory fine particulate matter (NR-PM) in Delhi. They have developed a "time of day" PMF method which improves on the traditional PMF algorithm primarily because it does not rely on the assumption of source chemical profiles being static over time. Instead, the authors use their new method to explore the possibility of diurnal and seasonal source profile changes in typical aerosol mass spec PMF factors such as BBOA, COA, and hydrocarbon-like organic aerosol (HOA). Most of the nuts and bolts of this method is covered in their companion paper, almost to the point that the AMT manuscript is an extended "methods" section of this manuscript. Naturally, the discussion in this manuscript is thus quite reliant on the methods presented in the AMT manuscript. Ideally,this manuscript should be considered for publication after the manuscript presenting the underlying methods has been peer-reviewed. As such, I will preface the remnant of my review with the following: this review of ACP manuscript assumes that the underlying methods presented in the companion AMT manuscript do not have any major technical issues. This manuscript focuses on how the authors interpret their results, their advantages in informing source apportionment, and policy implications. Overall, I think this is a very informative approach developed by the authors. The manuscript is very thoroughly prepared and the results are presented clearly. I have a few comments that I tihnk should improve the readability of the manuscript. A couple of general comments first:*

*1. Because several different PMF runs are being compared in this manuscript, I understand why the authors decided to tag each PMF run with the SYYTTTT code. But is the "YY" component really needed? As far as I can tell, all measurements occurred in 2017. I suggest removing the YY component and hyphenating to make the code more readily graspable i.e., S-TT-TT.*

Response: We agree with the reviewer that the YY component is not really needed. Thus, all figures, tables, and discussion in the main manuscript now refer to the corresponding runs using the notation suggested, S-TT-TT.

*2. I think the authors need to be cautious in claiming that all differences between their "time of day" PMF approach and the conventional PMF approach can be attributed SOLELY to potential changes in source chemical signatures. After all, PMF is blind to chemistry and it simply tries to find a local minima in Q/Qexp for whatever dataset is provided to it. Time-varying instrumental uncertainties, relative ionization efficiencies, etc., can also play a role in how this solution is found. In fact, for the same input dataset and same number of factors specified, multiple slightly different PMF solutions can be derived based on how the matrix is rotated ("f_peak"), or how the first numerical step is taken by the algorithm("seed"). Attributing*

*variations between PMF runs to source chemical signatures without exploring (or at least acknowledging) these other possible sources of variations seems risky.*

Response: We agree with the reviewer that the differences between simply conducting PMF by time-of-day versus conventional PMF could in theory be due to other factors that influence the statistical solution (for example, uncertainties and time variations in relative ionization efficiencies). We also agree that statistical uncertainty (random error and rotational ambiguity due to time-varying instrumental uncertainties) can play a role in the solutions as well. To account for the statistical sources of variations, we used the error estimation techniques available in the EPA PMF tool. These techniques systematically account for random error and rotational ambiguity (Sect. 2.5, Bhandari et al., AMTD, 2022) and we only present results that pass our QA/QC procedures (Sect. S1, Bhandari et al., AMTD, 2022). The limitations of the statistical solution of the time-of-day PMF approach (changing influence of outliers and number of zeros) are presented in our companion paper (Sect. 1, Bhandari et al., AMTD, 2022). We also demonstrate the mathematical proof of statistical improvement in fits based on the time-of-day PMF approach over the seasonal PMF approach in the companion paper (Sect. 2.2, Bhandari et al., AMTD, 2022).

To address the reviewer's comments, we have added the following text in Sect. 2.3 PMF (ME-2) analysis:

"Similar to the seasonal PMF approach typically adopted, the time-of-day PMF approach does not account for uncertainties and time variations in relative ionization efficiencies of the detected species (Xu et al., 2018; Katz et al., 2021). Additionally, the sensitivity of the time-of-day PMF approach to the changing influence of outliers and the number of zeros has not been evaluated (Sect. 1, Bhandari et al., 2022)."

*Specific comments:*

*L104: the term "NR-PM2.5" is used here for the first time. Please define. Also, if not already specified, please specify that the DAS ACSM measured NR-PM1, not NR-PM2.5.*

Response: We have specified that DAS ACSM measured NR-PM1 (Sect. 1). For the ease of the user, we have provided all abbreviations in Appendix A. We have also updated the text (Sect. 1):

"Based on NR-PM$_{2.5}$ (non-refractory PM smaller than 2.5 μm in diameter) Aerosol Chemical Speciation Monitor (ACSM) data collected, one study identified hydrocarbon-like organic aerosol (HOA) and solid-fuel combustion organic aerosol (SFC-OA) as the primary PMF factors, with HOA contributing ~15% and ~17%, and SFC-OA contributing ~25% and ~17% to total organic NR-PM$_{2.5}$ in winter 2018 and summer 2018 (the month of May), respectively (Tobler et al., 2020)."

*L118: "… its ubiquity". Please clarify what is being referred to here. Ubiquity of cooking sources? Or ubiquity of detectable COA? Its a hair-split, but good to clarify.*

Response: We were referring to the ubiquity of cooking sources. We have also updated the text (Sect. 1):

"Overall, online mass spectrometry studies have reported limited measurements of biomass burning contributions and have not always resolved cooking organic aerosol (COA) as a factor in Delhi, despite the ubiquity of cooking sources in this area (Tobler et al., 2020; Shukla et al., 2021; Lalchandani et al., 2021)."

*L135: "IGOR PET" – Igor is the software, which isn't an acronym, so shouldn't be capitalized. And PET is a super-acronym (the "P" stands for an acronym itself), which should be defined. Also, since PET is first mentioned here, the Ulbrich 2009 citation should be included here.*

Response: We have added PET as an acronym in Appendix A. We have also updated the text (Sect. 1):

"Then, we deployed the Igor PET (PMF Evaluation Tool) tool (Ulbrich et al., 2009) on seasonal datasets and 2–3 PMF factors were extracted (Bhandari et al., 2020)..."

*L164: at the end of "… course of the day", I suggested adding a clause "especially OOA factors", because the phenomena described later (reaction chemistry, gas-particle partitioning, changing meteorology, etc.) would affect the chemical signature of OOA factors more than primary factors.*

Response: We have updated the text (Sect. 2.1):

"A key limitation of PMF is that it assumes constant MS profiles, even though source signatures can change over the course of the day, especially for OOA factors."

*L240: If I am understanding Figure 1 correctly, the sentence "clearly, POA concentrations exhibit larger variability than OOA concentrations" should end with the phrase "in winter". Monsoon POA and OOA both seem to be similarly variable (0 to 50 ug/m3, from the looks of the y-scale).*

Response: We have updated the text (Sect. 3):

"Clearly, POA concentrations exhibit larger variability than OOA concentrations in winter."

*Figure 1: I suggest a few tweaks to this plot – a) add Spearman R to each panel comparing "time of day" to "seasonal" PMF; b) adding "WINTER" and "MONSOON" to the panel headers, and adding a dividing line between (a,b) and (c,d) will make the seasonal distinction jump out better.*

Response: We have updated Figure 1 according to the reviewer's suggestions.

[Figure]

L291 – 304: comparisons are made to other studies that made measurements at 3m height near an arterial road, but it would be help to also include the DAS study parameters here (I know 4th floor of the building, but just to compare numbers, please include sampling inlet height off of the ground, and distance to nearest major roadway).

Response: We have updated the text, also accounting for the comments of the other reviewer (Sect. 3.1.1):

"In winter, we observe average contributions of HOA as 16% and BBOA as 19% to total organic non-refractory submicron aerosols (NR-PM1) (and no detected contributions of cooking), in line with a recent online mass spectrometer deployment for NR-PM2.5 (Tobler et al., 2020: HOA ~15%, SFC-OA~25%; Lalchandani et al., 2021: HOA ~20%, SFC-OA~30%; Table S11). The sampling periods of these studies capture similar periods; the current study sampled between Jan 15-Feb 14 (2017), whereas Tobler et al (2020) sampled between Dec 22-Jan 17 (2018). The distance of the sampling sites to nearest major roadway are 150 m in present work and 50 m in the other study (Tobler et al., 2020), whereas the heights of sampling are 15 m and about 9 m (rooftop of a double story building), respectively. Also, the two sampling sites are within 10 km of each other. There are large differences of relative contributions in comparison to another study (Reyes-Villegas et al., 2021: HOA~40%, BBOA~12%, COA~8%). The sampling site in this study is at a distance of about 6 km from both the site of Tobler et al (2020) and the current study. Additionally, the study sampled between Feb 5-Mar 3 (2018), capturing the winter-spring transition rather than peak winter, leading to lower expected contributions of biomass burning for winter-based heating purposes. The study conducted ground-based measurements (~3 m high), within 50 m of a major road. The siting and the sampling period likely resulted in high contributions of HOA to OA in this study. The observation of similar or larger contributions of biomass burning than traffic to organics in the winter of Delhi is consistent with several filter-based receptor modeling studies as well (Chowdhury et al., 2007; Pant et al, 2015; IIT Kanpur, 2016; Sharma and Mandal, 2017; Jaiprakash et al., 2017). In monsoon, we observe ~14% organic NR-PM1 attributable to HOA and ~15% attributable to COA (no detected contributions of biomass burning) (Table S12). These results are in line with the online mass spectrometry study of Tobler et al. (2020; May 2018: NR-PM2.5 contributions of HOA (17%) and SFC-OA (17%)). The sampling periods of these studies capture somewhat different periods; the current study sampled between Jul 1-Sept 15 (2017), whereas Tobler et al (2020) sampled between May 1-May 26 (2018). However, there are large differences in comparison to another study, particularly in HOA contributions (Cash et al., 2021: HOA~30%, SFC-OA~11%, COA~6%). However, this study conducted ground-based measurements (~3m high), within 250m of multiple traffic sources (railways, major road). This sampling site is at a distance of about 7 km from the site of Tobler et al (2020) and about 10 km. This siting likely resulted in high contributions of HOA to OA in that study. Additionally, this study sampled between Aug 3-Aug 19, a relatively shorter sampling period in comparison to the other studies. Differences in sampling periods also lead to a difference in influencing meteorology, which could be contributing to the differences across these studies."

In addition, we have updated the text elsewhere as well (Sect. 2.2):

"As a part of the DAS campaign, an ACSM (Aerodyne Research, Billerica, MA) was operated at ~1 min time resolution in a temperature-controlled laboratory on the top floor of a four-story building (15 m) at IIT Delhi (Ng et al., 2011b). This sampling site is located about 150 m from an arterial roadway."

*L334: the nomenclature "winter-to-monsoon" sounds like a ratio of winter to monsoon levels. Instead, I suggest something like "monsoon-adjusted winter BBOA".*

Response: We have updated the text here and elsewhere in the manuscript, replacing the nomenclature "winter-to-monsoon" with "monsoon-adjusted winter" and the nomenclature "monsoon-to-winter" with "winter-adjusted monsoon".

*L345: please verify precip data shown in Fig S1. Seems like an abrupt step change at 6 am. If there is an explanation for this, please include in Fig S1 caption.*

Response: We agree with the reviewer regarding the discontinuity. On further analysis, we found the discontinuity persisting across seasons. Similar discontinuities have been reported elsewhere as well (ResearchGate, 2021). The discontinuities in precipitation data, retrieved from the European Centre for Medium-Range Weather Forecasts' reanalysis dataset, ERA-Interim (now, ERA5), have been attributed to the "initialization of the analysis windows", which are 12 hours each in length. Due to this, the analyses results vary smoothly within separate assimilation windows but discontinuities can occur where windows change. Different variables have different windows, and the summarized diurnal pattern in Fig. S1 suggests that the 12 hour windows for precipitation data are 0600-1800 LT and 1800-0600 LT. We have clarified this issue in the caption of Fig. S1, where we have added the following text:

"The ERA-Interim 12-hour long assimilation windows for precipitation data are from 0600-1800 LT and 1800-0600 LT. The discontinuities in precipitation data occur where windows change. Similar discontinuities have been reported elsewhere as well (ResearchGate, 2021)."

*Figs 2 and 3: I don't see why these two cannot be combined into one figure, similar to Figure 1. Irrespective of whether authors choose to combine Figs 2 and 3, please consider adding a text label to the figures clearly specifying the season.*

Response: We have updated Figures 2 and 3 with the text labels to the figures clearly specifying the season. We have retained them as separate figures due to figure density and different stories in the figures.

*L385: The studies cited here for "Asian cooking" references were not conducted on Indian cooking styles. Some of them were on Chinese cuisine emissions (or measurements conducted in China). I would be careful about bucketing Chinese and Indian cooking together as "Asian" here, without at least acknowledging that these two are vastly different and there just isn't literature on the latter as there is on the former.*

Response: We have updated the text (Sect. 3.2):

"A wide variety of cooking styles are in use in Delhi: from more regulated liquified petroleum gas connections to wood, dung, and waste burning using open stoves. These different cooking styles would result in different COA profiles. An accurate and detailed representation

of cooking organic aerosols in Delhi requires accounting for the variability in cooking fuels or technology in the Indian context using different characteristic markers. However, in our knowledge, no laboratory studies have been conducted on COA for Indian cooking styles. The COA MS for these cooking styles could have different mass spectral features than those used in this study which are based on studies elsewhere."

*L390 and several other places: the m/z 55:57 signal ratio is used often to infer cooking influence, but it is not clear what is a reference value for this ratio to compare against while making this inference. For example, L416: why is the 55:57 value of 1.1 "low"? This is where those reference values from Robinson et al. and Mohr et al. would be helpful. I suggest that when the Robinson 2018 study is cited first here, also include the 55:57 ratio from that study, so as to "set the stage" for upcoming discussion on 55:57 ratios. Also, Claudia Mohr's 2012 study ([https://acp.copernicus.org/articles/12/1649/2012/](https://acp.copernicus.org/articles/12/1649/2012/)) was an earlier one that used the 55:57 ratio to identify cooking emissions. It should be cited here along with Robinson, and example values of 55:57 ratios from both should be mentioned for reference.*

Response: We thank the reviewer for the additional reference of Mohr et al (2012). In this study, we used the Robinson et al (2018) ratio of contributions at m/z 55:57 of 1.6 as a preliminary test for relative positioning of the HOA and COA profiles (COA factors with the ratio closer or greater than 1.6 and HOA profiles with the ratio significantly lower than 1.6). Mohr et al (2012) fit HOA and COA spectra from previous source apportionment and source emission studies and provide estimates for the ratio of contributions at m/zs 55:57 as $3.0\pm0.7$ for COA and $0.9\pm0.2$ for HOA. While both studies observed large spreads and continuities in the ratio depending on whether the factor/plume was HOA-influenced or COA-influenced (Fig. 6, Mohr et al., 2012; Fig. S1, Robinson et al., 2018), both studies point to higher values of the ratio in the COA MS relative to the HOA MS. Indeed, we observe that the ratio in the COA MS is spread around 1.6 (min.:1.5, max.: 2.0) compared to the ratio in the HOA MS being spread around 1 (min.:0.8, max.: 1.2), a statistically significant difference (Tables S17, S18). We have updated the text (Sect. 3.2):

"The ratio of contributions at m/z 55 to m/z 57 has been identified as a marker for the presence of cooking organic aerosol (Mohr et al., 2012; Robinson et al., 2018). In this study, we used the Robinson et al (2018) ratio of contributions at m/z 55:57 of 1.6 as a preliminary test for relative positioning of the HOA and COA profiles (COA factors with the ratio closer or greater than 1.6 and HOA profiles with the ratio substantially lower than 1.6). Mohr et al (2012) fit HOA and COA spectra from previous source apportionment and source emission studies and provide estimates for the ratio of contributions at m/zs 55:57 as $3.0\pm0.7$ for COA and $0.9\pm0.2$ for HOA. While both studies observed large spreads and continuities in the ratio depending on whether the factor/plume was HOA-influenced or COA-influenced (Fig. 6, Mohr et al., 2012; Fig. S1, Robinson et al., 2018), both studies point to higher values of the ratio in the COA MS relative to the HOA MS."

*L653: "with larger ratios of contributions of m/z 55 to m/z 57" please include a number or range here to quantify what "larger" means.*

Response: We have updated the text (L652-654):

"Winter periods with cooking influence (1100–2300 hours) display POA MS profiles with larger ratio of contributions of m/z 55 to m/z 57 (1.14±0.04) in comparison to seasonal PMF POA MS profiles (1.02±0.02), in line with the strong cooking influence expected in these periods (Tables S19 and S20).

*References:*

1. *Bhandari, S., Arub, Z., Habib, G., Apte, J. S., and Ruiz, L. H.: Source apportionment resolved by time-of-day for improved deconvolution of primary source contributions to air pollution, https://amt.copernicus.org/preprints/amt-2022-76/, submitted to Atmospheric Measurement Techniques, 2022.*
2. *Katz, E. F.; Guo, H.; Campuzano-Jost, P.; Day, D. A.; Brown, W. L.; Boedicker, E.; Pothier, M.; Lunderberg, D. M.; Patel, S.; Patel, K.; Hayes, P. L.; Avery, A.; Hildebrandt Ruiz, L.; Goldstein, A. H.; Vance, M. E.; Farmer, D. K.; Jimenez, J. L.; DeCarlo, P. F. Quantification of Cooking Organic Aerosol in the Indoor Environment Using Aerodyne Aerosol Mass Spectrometers. Aerosol Science and Technology 2021, 55 (10), 1099–1114. https://doi.org/10.1080/02786826.2021.1931013.*
3. Mohr, C.; DeCarlo, P. F.; Heringa, M. F.; Chirico, R.; Slowik, J. G.; Richter, R.; Reche, C.; Alastuey, A.; Querol, X.; Seco, R.; Peñuelas, J.; Jiménez, J. L.; Crippa, M.; Zimmermann, R.; Baltensperger, U.; Prévôt, A. S. H. Identification and Quantification of Organic Aerosol from Cooking and Other Sources in Barcelona Using Aerosol Mass Spectrometer Data. *Atmospheric Chemistry and Physics* 2012, *12* (4), 1649–1665. https://doi.org/10.5194/acp-12-1649-2012.
4. ResearchGate, https://www.researchgate.net/post/Has-anybody-observed-strange-discontinuities-in-ERA5s-diurnal-cycle-of-temperature-precipitation-etc, 2021.
5. Robinson, E. S.; Gu, P.; Ye, Q.; Li, H. Z.; Shah, R. U.; Apte, J. S.; Robinson, A. L.; Presto, A. A. Restaurant Impacts on Outdoor Air Quality: Elevated Organic Aerosol Mass from Restaurant Cooking with Neighborhood-Scale Plume Extents. *Environ. Sci. Technol.* 2018, *52* (16), 9285–9294. https://doi.org/10.1021/acs.est.8b02654.
6. Xu, W.; Lambe, A.; Silva, P.; Hu, W.; Onasch, T.; Williams, L.; Croteau, P.; Zhang, X.; Renbaum-Wolff, L.; Fortner, E.; Jimenez, J. L.; Jayne, J.; Worsnop, D.; Canagaratna, M. Laboratory Evaluation of Species-Dependent Relative Ionization Efficiencies in the Aerodyne Aerosol Mass Spectrometer. *Aerosol Science and Technology* 2018, *52* (6), 626–641. https://doi.org/10.1080/02786826.2018.1439570.

**Reviewer 2**

*This work by Bhandari et al. utilizes their companion study (Bhandari et al., 2022 AMT Discussions) that proposes a time-resolved method for source apportionment using the underlying approach of positive matrix factorization (PMF), also referred as "time-of-day PMF" and demonstrated statistical improvements over the traditional PMF (uncertainty owing to static mass spectral profiles). Delhi, India is one of most polluted megacities on Earth, with inarguably one of the highest Primary Organic Aerosol (POA) concentrations anywhere. This study critically focusses on Delhi to quantify the contributions of different POA components: BBOA (biomass burning), COA (cooking), and hydrocarbon like organic aerosol (HOA, from anthropogenic fossil fuel combustion) by applying "time-of-day PMF" (diurnal profiles as a result) on two seasons (winter and monsoon 2017) using OA measurements from an Aerosol Chemical Speciation Monitor (ACSM). They utilize the EPA PMF tool with the underlying Multilinear Engine (ME-2) as the PMF solver, and conduct detailed uncertainty analysis for statistical validation of their results. Assuming that the companion AMT manuscript will eventually go through final publication without any technical modifications in its method (given it's the bulk of the "Method" section of this paper as well), I think this work is very significant for better design policies to mitigate pollution in Delhi or National Capital Region (NCR, in vicinity of Delhi) caused by relevant primary sources of organic aerosols as analyzed in detail with the time-resolved component in this study.*

*I will suggest publication of this work, after the following comments are addressed by the authors:*

*General comments:*

■ *Will suggest the authors to include updated citation of the finally published companion Bhandari et al., 2022 AMT paper as its it's the bulk of the underlying principle/Method of this section. That would be ideal before publishing this work. Any modifications/edits to the companion AMT paper on its final publication should be accommodated in current work (ideally before a final draft is accepted, if possible or as an addendum later)*

Response: We agree with the reviewer that the updated citation of the companion paper be included. We submitted these companion papers at the same time and have received positive reviews on the companion paper, and will be submitting our response to those reviews within the next day or two. Thus, the reviewers and editors will have the chance to review the reviewers comments and our responses before making a decision. Our edits to the companion paper are minor, and no changes were made to the methods.

■ *The current study needs to further decipher the identification of different markers for the presence of cooking organic aerosol (COA) based on the variability in cooking fuels or technology in Indian context (more regulated liquified petroleum gas connections vs*

*wood or residual burning using open stoves- also presents a pragmatic contrast within different COA sources in Delhi), which is currently missing in the current discussion (Lines 385-395). It's understandable if it is beyond the scope of current study, but should be in that case, mentioned as a limitation of the current study that needs further exploration.*

Response: We agree with the reviewer that the current work does not account for the variability in cooking fuels or technology in Indian context, and this is a limitation of this work. We have added the following text (Sect. 3.2):

"A wide variety of cooking styles are in use in Delhi: from more regulated liquified petroleum gas connections to wood, dung, and waste burning using open stoves. These different cooking styles would result in different COA profiles. An accurate and detailed representation of cooking organic aerosols in Delhi requires accounting for the variability in cooking fuels or technology in the Indian context using different characteristic markers. However, in our knowledge, no laboratory studies have been conducted on COA for Indian cooking styles. The COA MS for these cooking styles could have different mass spectral features than those used in this study which are based on studies elsewhere."

*Specific (minor) comments:*

*Line 70: Please refer '… and co-workers' as '… et al. (YEAR)' consistent with other instances in the manuscript text (rephrase Lines 70-72 accordingly, edit other such instances in the manuscript accordingly).*

Response: We have updated the text here and elsewhere in the manuscript, replacing the "… and co-workers" as "… et al. (YEAR)". We rephrased lines 70-72 accordingly as well (Sect. 1):

"In two companion studies, PMF was conducted on 50–60 species of organics collected on filter samples between Dec 2016–Dec 2017 representing five classes of polar organic compounds (Gadi et al., 2019; Shivani et al., 2019)."

*Lines 96-97: Add space between 'µg' and 'm-3' (applies to similar other instances in the manuscript text).*

Response: We have updated the text here and elsewhere in the manuscript, adding a space between 'µg' and 'm$^{-3}$'.

*Lines 99-101: Same point about citation being consistent: Rooney et (2019) and Rooney (2019) in consecutive lines, although it's the same reference. Try to keep citation of a paper consistent throughout the manuscript.*

Response: We have updated the citations here and elsewhere in the manuscript to keep citations consistent with the format "… et al. (YEAR)".

*Lines 104 and 133-134: Please clarify the full form of any abbreviation at its first use in the manuscript, i.e. non-refractory (NR) and ACSM in this case at Line 104 instead of Lines 133-134. Similarly, apply for any similar instances in the manuscript.*

Response: For the ease of the user, we have provided all abbreviations in Appendix A. We have also updated the text here and elsewhere to clarify the full form of any abbreviation at its first use in the manuscript.

*Line 185: Rephrase "have been described previously" to "have been described in previous literature".*

Response: We have updated the text (L184-187):

"This approach is the commonly used approach, and the reasons for the selection of the specific set of *m/z* values have been described in previous literature (Zhang et al., 2005)."

*Lines 291-304 (Section 3.1.1): More discussions/hypothesis and/or details are needed on why there exists inconsistency in HOA average contributions to OA compared with previous studies? For instance , is difference in meteorology in different years between different studies a factor as well besides the difference in profile of emission sources at site(s) between different studies?*

Response: We agree with the reviewer that differences in studies could occur due to differences in meteorology in different years, as well as the differences in profile of emission sources at the sites between studies. We have updated the text (Sect. 3.1.1):

"In winter, we observe average contributions of HOA as 16% and BBOA as 19% to total organic non-refractory submicron aerosols (NR-PM1) (and no detected contributions of cooking), in line with a recent online mass spectrometer deployment for NR-PM2.5 (Tobler et al., 2020: HOA ~15%, SFC-OA~25%; Lalchandani et al., 2021: HOA ~20%, SFC-OA~30%; Table S11). The sampling periods of these studies capture similar periods; the current study sampled between Jan 15-Feb 14 (2017), whereas Tobler et al (2020) sampled between Dec 22-Jan 17 (2018). The distance of the sampling sites to nearest major roadway are 150 m in present work and 50 m in the other study (Tobler et al., 2020), whereas the heights of sampling are 15 m and about 9 m (rooftop of a double story building), respectively. Also, the two sampling sites are within 10 km of each other. There are large differences of relative contributions in comparison to another study (Reyes-Villegas et al., 2021: HOA~40%, BBOA~12%, COA~8%). The sampling site in this study is at a distance of about 6 km from

both the site of Tobler et al (2020) and the current study. Additionally, the study sampled between Feb 5-Mar 3 (2018), capturing the winter-spring transition rather than peak winter, leading to lower expected contributions of biomass burning for winter-based heating purposes. This study conducted ground-based measurements (~3 m high), within 50 m of a major road. The siting and the sampling period likely resulted in high contributions of HOA to OA in this study. The observation of similar or larger contributions of biomass burning than traffic to organics in the winter of Delhi is consistent with several filter-based receptor modeling studies as well (Chowdhury et al., 2007; Pant et al, 2015; IIT Kanpur, 2016; Sharma and Mandal, 2017; Jaiprakash et al., 2017). In monsoon, we observe ~14% organic NR-PM1 attributable to HOA and ~15% attributable to COA (no detected contributions of biomass burning) (Table S12). These results are in line with the online mass spectrometry study of Tobler et al. (2020; May 2018: NR-PM2.5 contributions of HOA (17%) and SFC-OA (17%)). The sampling periods of these studies capture somewhat different periods; the current study sampled between Jul 1-Sept 15 (2017), whereas Tobler et al (2020) sampled between May 1-May 26 (2018). However, there are large differences in comparison to another study, particularly in HOA contributions (Cash et al., 2021: HOA~30%, SFC-OA~11%, COA~6%). However, this study conducted ground-based measurements (~3m high), within 250m of multiple traffic sources (railways, major road). This sampling site is at a distance of about 7 km from the site of Tobler et al (2020) and about 10 km. This siting likely resulted in high contributions of HOA to OA in that study. Additionally, this study sampled between Aug 3-Aug 19, a relatively shorter sampling period in comparison to the other studies. Differences in sampling periods also lead to a difference in influencing meteorology, which could be contributing to the differences across these studies."

In addition, we have updated the text elsewhere as well (Sect. 2.2):

"As a part of the DAS campaign, an ACSM (Aerodyne Research, Billerica, MA, Ng et al., 2011b) was operated at ~1 min time resolution in a temperature-controlled laboratory on the top floor of a four-story building (15 m) at IIT Delhi.This sampling site is located about 150 m from an arterial roadway."

*Line 351: "relatively high volatility of BBOA": oxidized BBOA is low-volatility OA and more explanation needed here on why monsoons won't exhibit much of low-volatility oxidized BBOA?*

Response: We agree with the reviewer that oxidized BBOA can have low volatility. In this paper, we separate oxidized organic aerosol (OOA) factors along with primary organic aerosol (POA) factors, and the biomass burning influence on OOA MS is discussed in Sect. 3.3.2. Figure 2, however, discusses time series patterns of primary biomass burning concentrations (referred to as BBOA in this paper). In Fig. 2, we show that monsoon-adjusted winter BBOA concentrations (adjusted for the warmer temperatures and ventilation coefficient) have very low concentrations. To conduct this analysis, we used the volatility basis set approach (Sect.

S4). To be clearer in the discussion that we are talking about primary BBOA, we have updated the text (Sect. 3.1.3):

"Our results suggest that the inability to separate primary BBOA in monsoon, particularly in the middle of the day (0900–1700 hours), can be attributed to the volatility of primary BBOA, as can be seen from the low BBOA concentrations estimated from hybrid MLR-PMF and monsoon-adjusted winter BBOA concentrations. While monsoon-adjusted winter BBOA concentrations are large at other hours, those BBOA concentrations are likely associated exclusively with winter night-time space heating. Thus, the lack of BBOA concentrations in monsoon are due to absence of sources (e.g. residential heating) as well as the relatively high volatility of primary BBOA, which would result in near-zero concentrations during monsoon daytime temperatures."

Elsewhere, we have updated the text to reflect that the abbreviations HOA, BBOA, and COA refer to primary HOA, BBOA, and COA respectively (Sect. 2.4):

"Hereafter, we refer to the approach as "hybrid MLR-PMF" . We refer to the primary components as HOA, BBOA and COA, and their corresponding oxidized components as oxidized HOA, oxidized BBOA, and oxidized COA."

*Figs 2 and 3: Add "Monsoon 2017" and "Winter 2017" labels respectively to Figs. 2 and 3. (Applies to other figures also)*

Response: We have updated all figures in the manuscript with the text labels to the figures clearly specifying the season. We do not use the year since all measurements presented in the work were conducted in 2017.

*Section 3.1.3: more clarity is needed on the rationale or necessity of doing "Winter-to-Monsoon" and "Monsoon-to-Winter" weighing on diurnal PMF*

Response: In this work, supplemented by the hybrid MLR-PMF approach, we separated COA (primary cooking organic aerosol) only in monsoon, and BBOA (primary biomass burning organic aerosol) only in winter. This was a surprising observation: cooking is a ubiquitous activity performed across seasons, and the use of biomass burning in Delhi across seasons has been well-documented (Fu et al., 2010; Yadav et al., 2013; Pant et al., 2015, Rooney et al., 2019). By calculating "Winter-to-Monsoon" (now renamed as monsoon-adjusted winter) and "Monsoon-to-Winter" (now renamed winter-adjusted monsoon) weighing on diurnal PMF, we wanted to address fundamental questions along the lines of "how can cooking not be a source in winter?" and "is biomass burning really used less in monsoon?" The intercomparison across seasons gives leading evidence to address these questions: cooking aerosols are likely a very small fraction of primary aerosols in winter, and primary biomass burning emissions are likely too volatile to be in the particulate phase in summer. Finally,

obtaining factors of different mass spectral characteristics in different seasons also makes designing policies and implementation strategies to address air pollution tedious, which can make such policies harder to justify. We have updated the text to better reflect the importance of this analysis (Sect. 3.1.3):

"In this work, supplemented by the hybrid MLR-PMF approach, we separated COA (primary cooking organic aerosol) only in monsoon, and BBOA (primary biomass burning organic aerosol) only in winter (Sect. 3.1). This was a surprising observation—cooking is a ubiquitous activity performed across seasons, and the use of biomass burning in Delhi across seasons has been well-documented (Fu et al., 2010; Yadav et al., 2013; Pant et al., 2015, Rooney et al., 2019). To better understand the absence of COA in winter and BBOA in monsoon, we plot diurnal patterns of average factor concentrations in the two seasons in Figs. 2 and 3. We conduct an inter-seasonal comparison on this diurnal data. We use the 1-D volatility basis set (VBS) approach to adjust for temperature (T) and ventilation coefficient (VC). Details of this analysis are in the Supplement (Sect. S4). The intercomparison exercise suggests that the lack of separation of cooking aerosols in winter is due to their small contribution to primary aerosols, and the lack of separation of primary biomass burning in monsoon is because these emissions are likely too volatile to be in the particulate phase during this warm season."

*Figures 4, 7 and 10: Author(s) should consider combining/rearranging parts of Figures 4,7 and 10 based on if they are for Winter 2017 and Monsoon 2017. Also discussions pertaining to these figures in Section 3 can also be further synthesized to improve readability.*

Response: The focus of the paper is on primary organic aerosols in Delhi, and therefore Figs. 4 and 7 are currently separated corresponding to the winter and monsoon primary aerosols respectively. We also saw interesting observations related to oxidized organics, that we discuss in a limited fashion, given the focus on primary organics. Currently, Fig. 10 discusses total OOA of the two seasons together. In the revised manuscript, we have removed Fig. 10 as a separate figure and instead included OOA sub-figures corresponding to the two seasons in panels of Figs. 4 and 7.

*References:*

7. *Fu, P. Q., Kawamura, K., Pavuluri, C. M., Swaminathan, T., and Chen, J.: Molecular characterization of urban organic aerosol in tropical India: contributions of primary emissions and secondary photooxidation, Atmospheric Chemistry and Physics, 10, 2663–2689, URL https://acp.copernicus.org/articles/10/2663/2010/, 2010.*
8. Pant, P., Shukla, A., Kohl, S. D., Chow, J. C., Watson, J. G., and Harrison, R. M.: Characterization of ambient PM2.5 at a pollution hotspot in New Delhi, India and inference of sources, Atmospheric Environment, 109, 178–189, URL https://doi.org/10.1016/j.atmosenv.2015.02.074, 2015.

9. Rooney, B., Zhao, R., Wang, Y., Bates, K. H., Pillarisetti, A., Sharma, S., Kundu, S., Bond, T. C., Lam, N. L., Ozaltun, B., Xu, L., Goel, V., Fleming, L. T., Weltman, R., Meinardi, S., Blake, D. R., Nizkorodov, S. A., Edwards, R. D., Yadav, A., Arora, N. K., Smith, K. R., and Seinfeld, J. H.: Impacts of household sources on air pollution at village and regional scales in India, Atmospheric Chemistry and Physics, 19, 7719–7742, URL https://acp.copernicus.org/articles/19/7719/2019/, 2019.

10. Yadav, S., Tandon, A., and Attri, A. K.: Characterization of aerosol associated non-polar organic compounds using TD-GC-MS: a four year study from Delhi, India, Journal of Hazardous Materials, 252-253, 29–44, URL https://doi.org/10.1016/j.jhazmat.2013.02.024, 2013.